# Integrative transcriptome-wide analysis of atopic dermatitis for drug repositioning

Jaeseung Song [1], Daeun Kim [1], Sora Lee[1], Junghyun Jung[1,3], Jong Wha J. Joo[2] & Wonhee Jang [1✉]

Atopic dermatitis (AD) is one of the most common inflammatory skin diseases, which significantly impact the quality of life. Transcriptome-wide association study (TWAS) was conducted to estimate both transcriptomic and genomic features of AD and detected significant associations between 31 expression quantitative loci and 25 genes. Our results replicated well-known genetic markers for AD, as well as 4 novel associated genes. Next, transcriptome meta-analysis was conducted with 5 studies retrieved from public databases and identified 5 additional novel susceptibility genes for AD. Applying the connectivity map to the results from TWAS and meta-analysis, robustly enriched perturbations were identified and their chemical or functional properties were analyzed. Here, we report the first research on integrative approaches for an AD, combining TWAS and transcriptome meta-analysis. Together, our findings could provide a comprehensive understanding of the pathophysiologic mechanisms of AD and suggest potential drug candidates as alternative treatment options.

[1] Department of Life Sciences, Dongguk University-Seoul, 04620 Seoul, Republic of Korea. [2] Department of Computer Science and Engineering, Dongguk University-Seoul, 04620 Seoul, Republic of Korea. [3] Present address: Department of Clinical Pharmacy, School of Pharmacy, University of Southern California, 1985 Zonal Avenue, Los Angeles, CA 90089, USA. ✉email: wany@dongguk.edu

Atopic dermatitis (AD) is one of the most common chronic dermatological diseases. The prevalence of AD reported in children worldwide in 2019 was 10–20% and is increasing[1,2]. AD is characterized by skin lesion and pruritus, which is not life-threatening but severely affects the quality of life. It is sometimes accompanied by thyroid autoimmunities, mental health problems, and cancerous diseases with/without infectious complications[3–5]. Currently, monoclonal antibodies are used to treat severe AD, while topical steroids and antihistamines are the first-line treatment for mild-to-moderate AD[6]. However, long-term use of topical steroids or antihistamines can cause unwanted side-effects such as skin thinning, melanocyte inhibition, and gastrointestinal effects[7,8]. Therefore, alternate strategies for treating mild-to-moderate AD are necessary.

Several genetic risk factors or causal genes for AD have been identified by functional and computational studies[9,10]. Genetic variants associated with *filaggrin* (*FLG*), *ovo-like transcriptional repressor 1* (*OVOL1*), and *interleukin 6 receptor* (*IL6R*) were suggested as risk loci for AD by a multi-ancestry genome-wide association study (GWAS)[9]. Other functional or clinical studies suggested *IL-4*, *IL-13*, *toll-like receptor 2* (*TLR2*), *matrix metalloproteinase 9* (*MMP9*), and *MMP10* as susceptibility genes for AD[10,11]. However, the underlying mechanisms of AD pathogenesis have not yet been elucidated.

Since general GWAS utilizes large-scale genotype data to identify genetic variants that influence disease pathogenesis, the method is less optimized for interpreting multiple gene expression changes caused by variants in non-coding regions. Recently, transcriptome-wide association study (TWAS) was suggested as an improved approach to implement gene expression imputation using GWAS results for better interpretation[12,13]. TWAS predicts the gene expression levels of phenotypes by combining genotypes and gene expression weights calculated using *cis*-expression quantitative trait loci (eQTLs) with multiple prediction models. TWAS has provided new insights into the underlying genetic/transcriptomic mechanisms of several diseases and phenotypes, including Alzheimer's disease, pancreatic cancer, and neutrophil development[14–16].

We conducted TWAS using the largest up-to-date AD GWAS dataset obtained from a European population. Transcriptome meta-analysis with microarray and RNA sequencing (RNA-seq) datasets were performed to identify gene expression changes that could not be explained solely by the genetic backbone. The connectivity between gene expression signatures from TWAS and transcriptome meta-analysis was assessed by network analysis. Finally, we performed in silico drug repositioning by combining the results from TWAS and meta-analysis to identify alternative therapeutic options to treat AD. To the best of our knowledge, this is the first integrative analysis on AD to combine TWAS and meta-analysis. We believe that our results can help expand knowledge of the biological mechanisms of AD pathogenesis and the development of the therapeutic options for AD.

## Results

### Enrichment analysis of GWAS signals from AD GWAS summary statistics.
To examine the genetic landscape of AD, this study uses the UK Biobank GWAS data consisting of 279,476 controls and 9831 AD patients. First, we examined whether the GWAS signals for AD were specifically enriched in certain tissue or cell types by using the functional mapping and annotation of genetic association (FUMA). We found that the *cis*-regulated genes of GWAS signals were mainly over-expressed in skin tissues (Supplementary Fig. S1)[17]. Next, tissue- or cell-specific heritability was analyzed using a linkage disequilibrium (LD) score regression applied to specifically expressed genes (LDSC-SEG) using the

multi-tissue expression dataset and multi-tissue chromatin dataset following Finucane et al.[18]. Heritability of AD GWAS signals on the multi-tissue expression data showed significant enrichment (false discovery rate (FDR) < 0.05) in the blood and immune-related tissues (Supplementary Fig. S2a; Supplementary Data 1) and this pattern was replicated in the multi-tissue chromatin dataset (Supplementary Fig. S2b; Supplementary Data 2).

### Transcriptome-wide associations for AD.
To identify susceptibility genes for AD, we performed TWAS with functional summary-based imputation (FUSION), using eQTL panels from nine tissues that can cover the systemic features of AD. The tissue panels were skin-sun exposed, skin-not sun exposed, cells-transformed fibroblast, spleen, thyroid, whole blood, cells-Epstein–Barr virus (EBV)-transformed lymphocytes, Netherlands Twin Registry (NTR) blood, and Young Finns Study (YFS) blood panel. Among the total of 52,860 associations, 25 genes in 31 loci remained statistically significant after using a Bonferroni-corrected threshold ($P < 0.05$/number of associations $(52,860) = \sim 9.46 \times 10^{-7}$) (Fig. 1a, Table 1, and Supplementary Data 3). Although TWAS signals showed the highest mean effect size in the skin-not sun-exposed panel, this was not dramatically higher than the mean effect sizes of other panels, indicating that the genetic features of AD may evenly affect the gene expression levels of nine tissue panels (Supplementary Fig. S3). The numbers of significant associations were six in skin-sun exposed, five in skin-not sun exposed, five in cells-transformed fibroblast, one in spleen, seven in thyroid, eight in whole blood, one in cells-EBV-transformed lymphocytes, two in NTR blood, and three in YFS blood panel. These results may represent the tissue-specific genetic features of AD in skin functions, immunological abnormalities, and thyroid autoimmunity.

Among these genes, 18 well-known AD risk genes such as *FLG*, *OVOL1*, and *IL6R* were significantly associated with TWAS signals for AD, confirming the validity of our methods. We identified three non-coding RNAs significantly associated with AD (*AC007278.2*, *AC007248.7*, and *RP11-85K15.2*) and four novel AD genetic risk genes, *leucine rich repeat and Ig domain containing 4* (*LINGO4*), *regulatory factor X5* (*RFX5*), *prolyl-4 hydroxylase subunit alpha 2* (*P4HA2*), and *RNA binding motif protein 17* (*RBM17*), which were not identified in previous GWAS studies. Among the 25 significantly associated TWAS genes, the majority (76%), including previously reported and novel TWAS genes, remained statistically significant after the permutation test ($P < 0.05$), suggesting that our TWAS genes are statistically robust findings.

Then, we compared the TWAS results with two other gene prioritization methods: the multi-marker analysis of genomic annotation (MAGMA) and the COLOC method[19,20]. While MAGMA analyzes the associated genes based on their chromosomal positions, COLOC is an R package for analyzing colocalization events to calculate posterior probabilities (PP) for hypotheses 0–4 ($H_0$–$H_4$). We detected 68 genes significantly associated with AD using MAGMA by applying a Bonferroni-corrected threshold ($P < 2.64 \times 10^{-6}$) that overlapped with 12 genes from TWAS (Supplementary Fig. S4a). The COLOC results showed 27 colocalized signals for AD (PP3 + PP4 > 0.8 and PP4/PP3 > 2), among which more than half (15/27) were also prioritized in TWAS (Supplementary Fig. S4b, c). Among the 27 genes from COLOC, 13 overlapped with the results from MAGMA (Supplementary Fig. S4c). Nine genes were prioritized with all three methods: *OVOL1*, *ARFRP1*, *PPP2R3C*, *FAM177A1*, *CLEC16A*, *SLC2A4RG*, *ZBTB46*, *IL6R*, and *IL18RAP* (Supplementary Fig. S4c).

To analyze whether novel TWAS genes were jointly associated with AD, a conditional and joint analysis using FUSION was

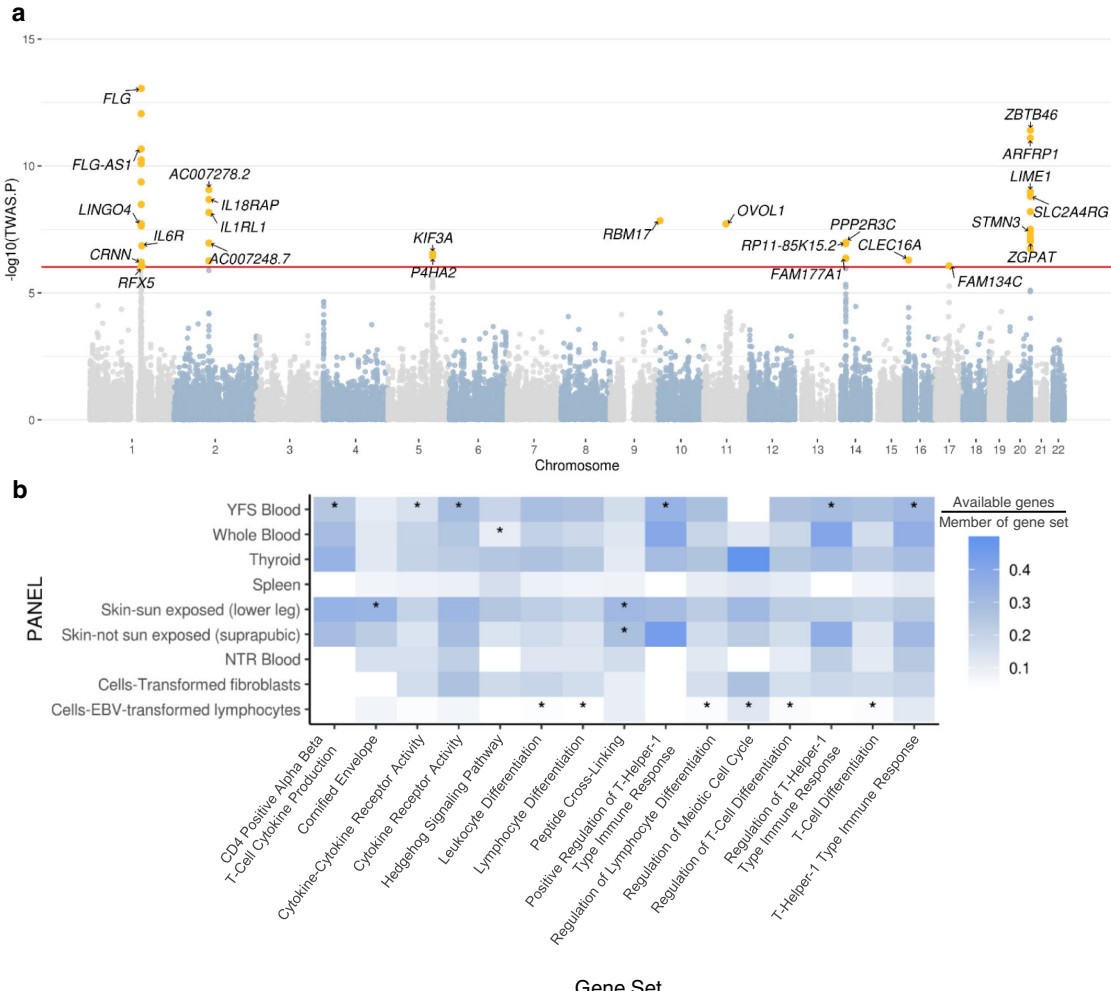

**Fig. 1 Overall results from the TWAS and post-analysis. a** A Manhattan plot showing the TWAS results obtained using the FUSION software. The red line indicates a Bonferroni-corrected threshold ($P < 9.46 \times 10^{-7}$), and the yellow dots correspond to the 25 TWAS-significant genes. **b** A heatmap showing the result of TWAS-GSEA. The color of each cell indicates the number of available genes involved in the gene set divided by the total number of the genes in the gene set. The cells marked with asterisks are the significantly enriched gene sets in the corresponding tissue panels.

conducted with the TWAS results (Supplementary Fig. S5a–c and Table 2). Among the four novel genes, *LINGO4*, *RFX5*, and *RBM17* remained jointly significant after the expected gene expressions were removed. A subsequent analysis using the fine-mapping of causal gene sets (FOCUS) was performed to determine the genetic causality of three novel jointly significant genes in AD pathogenesis. Two novel genes, *LINGO4* and *RBM17*, were included in credible sets with significant cross-validation *P*-values ($P < 0.05$) in FOCUS and their posterior inclusion probabilities (PIPs) indicating the nominal probability of causality were calculated (Supplementary Fig. S6a, b and Table 3). *LINGO4* was significantly detected in two genotype-tissue expression (GTEx) tissue panels: skin-sun exposed (PIP = 0.163) and skin-not sun exposed (PIP = 1). *RBM17* was also significantly detected in the skin-sun exposed panel (PIP = 0.695).

Overall TWAS signals were analyzed with TWAS-gene set enrichment analysis (TWAS-GSEA) software to determine their enriched biological pathway. Fifteen gene sets among the Gene Ontology–Biological Process (GO-BP) and Kyoto Encyclopedia of Genes and Genomes (KEGG) gene sets were significantly enriched with TWAS signals across five tissue panels: skin-sun exposed, skin-not sun exposed, YFS blood, whole blood, and cells-EBV-transformed lymphocytes (Fig. 1b and Supplementary

Data 4). TWAS signals were enriched in cornified envelope and peptide cross-linking in skin panels, which are well-known representative molecular characteristics of AD. TWAS signals from YFS blood and whole blood panels were significantly enriched in cytokine production (type 1 helper T cell activation) and hedgehog signaling pathways, which supports the notion that T cell-mediated immune responses are crucial pathogenic mechanisms of AD. In addition, we identified significant enrichment in TWAS signals in immune cell differentiation and meiotic cell cycle regulation from the cells-EBV-transformed lymphocytes panel. Together, the functional annotation of TWAS signals suggested that they mostly contribute to the abnormal activation of immune responses and the development of AD skin lesions.

**Transcriptome meta-analysis for AD**. Due to the complicated nature of AD, there may be transcriptional changes that can be marginally explained by genetic variations. Therefore, we conducted transcriptome meta-analysis to find transcriptional changes occurred by non-genetic factors. We collected skin transcriptome datasets (control: 93; AD: 140) from five studies on five different experiment platforms from public databases (Table 4). Then, we integrated the datasets into a merged set, removing the batch effects between individual studies. Principal

**Table 1 List of significantly associated genes from TWAS.**

| Chromosome | Gene | Panel | eQTL.ID | Z (TWAS) | P (TWAS) | P (Permutation) |
|---|---|---|---|---|---|---|
| | CRNN | Skin - not sun exposed (suprapubic) | rs4845763 | 4.9853 | 6.19E−07 | 0.1875 |
| | FLG | Cells - transformed fibroblasts | rs1552991 | −6.244 | 4.27E−10 | 0.0498 |
| | | Skin - sun exposed (lower leg) | rs11204948 | −7.4583 | 8.77E−14 | 0.0325 |
| | | Thyroid | rs4845737 | −7.1502 | 8.67E−13 | 0.0194 |
| | | Skin - not sun exposed (suprapubic) | rs1552991 | −6.6973 | 2.12E−11 | 0.0328 |
| | | Cells - transformed fibroblasts | rs1552991 | −6.5511 | 5.71E−11 | 0.0188 |
| 1 | FLG-AS1 | Skin - sun exposed (lower leg) | rs4845743 | −5.9174 | 3.27E−09 | 0.0335 |
| | | Spleen | rs4845737 | −5.5863 | 2.32E−08 | 0.0822 |
| | | Thyroid | | −6.4987 | 8.10E−11 | 0.0353 |
| | IL6R | Whole blood | rs4845618 | −4.9217 | 8.58E−07 | 0.0152 |
| | | YFS blood | rs4845623 | −5.2654 | 1.40E−07 | 0.0108 |
| | LINGO4* | Skin - not sun exposed (suprapubic) | rs12128071 | 5.6211 | 1.90E−08 | 0.0307 |
| | RFX5* | Thyroid | rs6684085 | −4.9048 | 9.35E−07 | 0.2308 |
| 2 | AC007278.2 | Whole blood | rs1420106 | −6.14 | 8.50E−10 | 0.0004 |
| | AC007248.7 | Whole blood | rs13015714 | 5.31 | 1.09E−07 | 0.0011 |
| | IL1RL1 | NTR blood | rs7559479 | −5.7959 | 6.80E−09 | 0.0123 |
| | IL18RAP | Whole blood | rs3755267 | −5.99 | 2.09E−09 | 0.0015 |
| | | YFS blood | rs3755266 | −5.0088 | 5.48E−07 | 0.0085 |
| 5 | KIF3A | Skin - sun exposed (lower leg) | rs3213639 | −5.1391 | 2.76E−07 | 0.1111 |
| | P4HA2* | Cells - transformed fibroblasts | rs4705928 | −5.085 | 3.68E−07 | 0.4444 |
| 10 | RBM17* | Skin - sun exposed (lower leg) | rs8463 | −5.6682 | 1.44E−08 | 0.005 |
| 11 | OVOL1 | Cells - EBV-transformed lymphocytes | rs10791824 | −5.6193 | 1.92E−08 | 0.001 |
| 14 | FAM177A1 | Skin - not sun exposed (suprapubic) | rs11156875 | 5.0543 | 4.32E−07 | 0.0157 |
| | PPP2R3C | NTR blood | rs8014377 | −5.3214 | 1.03E−07 | 0.0066 |
| | RP11-85K15.2 | Whole blood | rs13379372 | 5.3018 | 1.15E−07 | 0.0239 |
| 16 | CLEC16A | Thyroid | rs2286975 | 5.0211 | 5.14E−07 | 0.0478 |
| 17 | FAM134C | Skin - not sun exposed (suprapubic) | rs2293158 | −4.9214 | 8.59E−07 | 0.0026 |
| 20 | ARFRP1 | Cells - transformed fibroblasts | rs4809330 | 6.8397 | 7.93E−12 | 0.0019 |
| | | Thyroid | rs2315008 | 5.8079 | 6.33E−09 | 0.0082 |
| | | Whole blood | rs6062504 | −5.5349 | 3.11E−08 | 0.0062 |
| | LIME1 | Skin - sun exposed (lower leg) | rs6011040 | −5.2138 | 1.85E−07 | 0.1304 |
| | | Whole blood | rs4809330 | −5.3781 | 7.53E−08 | 0.0227 |
| | | YFS blood | rs6011058 | −6.1045 | 1.03E−09 | 0.0064 |
| | SLC2A4RG | Thyroid | rs1151622 | 6.0395 | 1.55E−09 | 0.0197 |
| | STMN3 | Skin - sun exposed (lower leg) | rs2315008 | −5.5004 | 3.79E−08 | 0.0102 |
| | | Whole blood | rs6011040 | 5.43 | 5.64E−08 | 0.0093 |
| | ZBTB46 | Thyroid | rs2315654 | 6.9417 | 3.87E−12 | 0.0015 |
| | ZGPAT | Cells - transformed fibroblasts | rs1058319 | 5.3406 | 9.26E−08 | 0.0086 |

Genes marked with asterisks are novel genes that were not identified in the original GWAS study.

**Table 2 Conditional and joint analysis results of novel TWAS genes in FUSION.**

| Gene | Z (TWAS) | P (TWAS) | Z (Joint) | P (Joint) | Tissue |
|---|---|---|---|---|---|
| LINGO4 | 5.6 | 1.90E−08 | 5.6 | 1.90E−08 | Skin - not sun exposed (suprapubic) |
| RFX5 | −4.9 | 9.40E−07 | −4.9 | 9.40E−07 | Thyroid |
| RBM17 | −5.7 | 1.40E−08 | −5.7 | 1.40E−08 | Skin - sun exposed (lower leg) |

Only jointly significant genes are displayed. Z (TWAS) and P (TWAS) are the original z-statistics and P-values from TWAS, respectively. Z (Joint) and P (Joint) are the z-statistics and P-values after conditioning on the TWAS signals, respectively.

**Table 3 Fine-mapping results of the novel TWAS genes using FOCUS.**

| Gene | Tissue | Chromosome | Model | P (Cross-validation) | PIP | Region |
|---|---|---|---|---|---|---|
| LINGO4 | Skin - sun exposed (lower leg) | 1 | enet | 0 | 0.163 | 1:148512062-1:151538786 |
| | Skin - not sun exposed (suprapubic) | | lasso | 0 | 1 | 1:151539165-1:153180729 |
| RBM17 | Skin - sun exposed (lower leg) | 10 | lasso | 0.0399 | 0.695 | 10:5983762-10:7171183 |

Only significant genes are displayed.

 ARTICLE

**Table 4 List of the transcriptome datasets used for transcriptome meta-analysis.**

| ID | Title | Control | Disease | Total | Source | Platform |
|---|---|---|---|---|---|---|
| GSE121212 | Atopic Dermatitis, Psoriasis and healthy control RNA-seq cohort[101] | 38 | 27 | 65 | Skin | Illumina HiSeq 2500 |
| GSE16161 | Broad defects in epidermal cornification in atopic dermatitis (AD) identified through genomic analysis[102] | 9 | 9 | 18 | Skin | Affymetrix Human Genome U133 Plus 2.0 |
| GSE5667 | Transcription data from Normal Skin and Nonlesional and Lesional Atopic Dermatitis/Eczema Skin[103,104] | 5 | 6 | 11 | Skin | Affymetrix Human Genome U133A Array Affymetrix Human Genome U133B Array |
| GSE120721 | Identification of novel immune and barrier genes in atopic dermatitis by means of laser capture microdissection[105] | 22 | 15 | 37 | Skin | Affymetrix Human Genome U133 Plus 2.0 |
| E-MTAB-8149 | Microarray transcriptome profiling of atopic dermatitis and psoriasis patients compared to healthy volunteers[106] | 19 | 83 | 102 | Skin | Affymetrix Human Gene 2.1 ST Array |
| Total | | 93 | 140 | 233 | | |

**Table 5 List of the novel genes from the transcriptome meta-analysis.**

| Gene | Entrez ID | log$_2$FC | P | FDR |
|---|---|---|---|---|
| C1orf162 | 128346 | 1.011 | 3.94E−53 | 8.68E−52 |
| NOCT | 25819 | 1.0662 | 1.60E−81 | 2.87E−79 |
| TIGAR | 57103 | 1.1068 | 1.67E−70 | 1.17E−68 |
| SCIN | 85477 | −1.2283 | 5.28E−85 | 1.37E−82 |
| BOC | 91653 | −1.0288 | 2.01E−106 | 2.19E−103 |

component analysis (PCA) was conducted to verify that major variances between samples were mainly due to disease state (Fig. 2a).

A transcriptome meta-analysis for identifying differentially expressed genes (DEGs) between AD and control groups was conducted using the batch effect-corrected merged set. Using merging data, we obtained robust genetic features (meta-signatures) with increased statistical power. We identified 268 meta-signatures consisting of 196 up- and 72 downregulated DEGs (FDR < 0.01 and |log$_2$fold-change (FC)|values > 1). We found that 226 genes from meta-signatures were included in at least one of the single datasets, while 42 were only identified in the meta-analysis (Fig. 2b). There was a clear distinction of gene expression profiles between the control and AD samples (Fig. 2c).

Among 268 meta-signatures, we identified five novel genes not previously reported as having associations with AD pathogenesis (Table 5). *Chromosome 1 open reading frame 162 (C1orf162)* was detected as a positively regulated gene and expresses a protein located in the hydrophobic region of the cellular membrane[21,22]. *Nocturnin (NOCT)* encodes a protein that is crucial in the circadian system[23]. The multi-functioning gene *TP53-induced glycolysis regulatory phosphatase (TIGAR)*, known for its role in the p53/TIGAR signaling pathway, was also significantly upregulated[24]. There were two downregulated novel genes: *scinderin (SCIN)* and *BOC cell adhesion associated, oncogene regulated (BOC)*. *SCIN* is associated with skin development or epithelial–mesenchymal transition, whereas *BOC* is involved in developmental pathways such as hedgehog pathway or neuronal differentiation[25–28].

We examined the expression profiles of TWAS genes in the meta-analysis results. Among the 25 TWAS genes, 16 had corresponding probes available in our merged set. Only *FLG* was involved in both the TWAS signal and meta-signature. Other TWAS genes, except *RBM17* (FDR = 0.258, log$_2$FC = 0.022), showed marginally significant differential expression (FDR < 0.01, |log$_2$FC| > 0) in our meta-analysis (Supplementary Table S1). Although there was only one direct overlap between TWAS genes and meta-signatures, we observed significant correlations between the two in gene-set levels (Supplementary Fig. S7). In line with the significant enrichment of TWAS results in meta-signatures, the functional enrichment results of the meta-analysis well conformed with the TWAS-GSEA results. We found that 80% of gene sets that were significantly enriched with TWAS signals were also enriched with the pre-ranked gene lists generated using the transcriptome meta-analysis (Supplementary Data 5). Together, the meta-analysis using published transcriptome data showed the reliability of the TWAS genes and identified five novel genes.

**Network construction and sub-network analysis for integrating TWAS and meta-analysis**. To systematically assess the connections between TWAS genes and meta-signatures, we conducted network analysis using both sets of genes as input nodes in the search tool for the retrieval of interacting genes (STRING) database (Supplementary Fig. S8a). After constructing

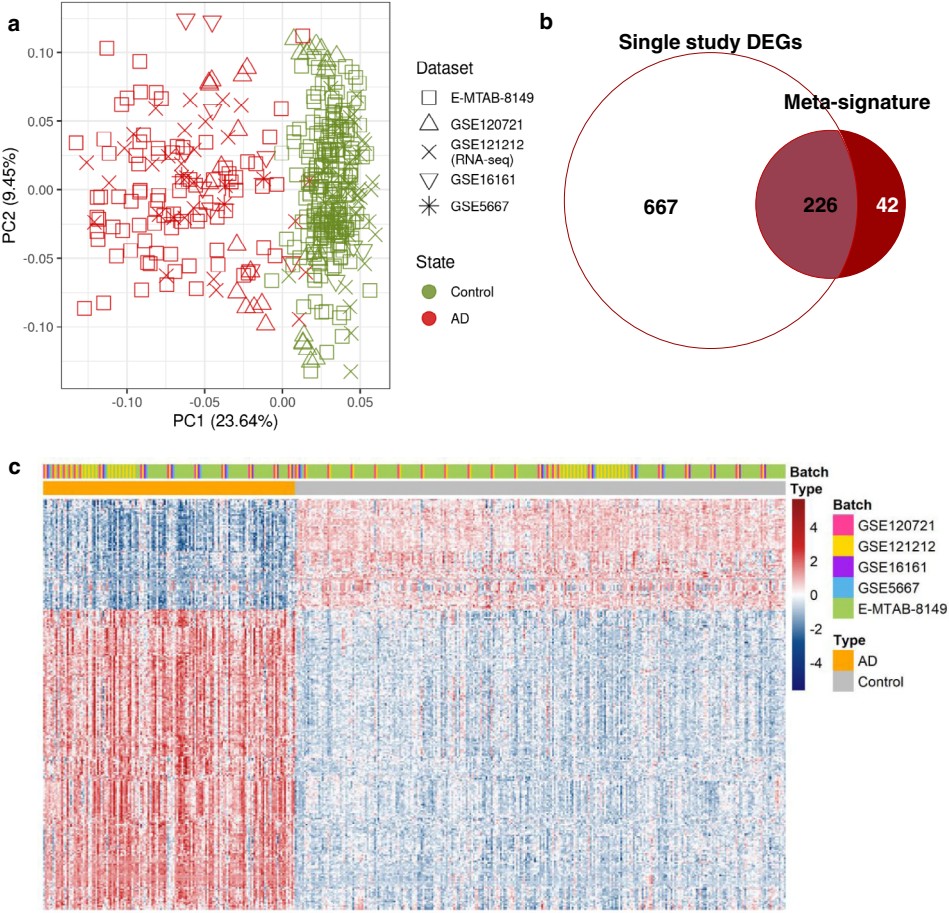

**Fig. 2 Correction of batch effects and identification of meta-signatures for AD. a** A scatter plot displaying the PCA results using all genes after the batch effect correction. The shapes of the points indicate the samples from each dataset. Green and red color correspond to the healthy control samples and AD samples, respectively. **b** A Venn-diagram comparing the DEGs from single studies with meta-signatures. **c** A heatmap of expression profiles of meta-signatures across the samples.

protein–protein interaction (PPI) networks composed of 243 nodes, we analyzed the sub-network clusters to examine the local connections between TWAS genes and meta-signatures. Networks were clustered into 12 sub-networks, and the three clusters with the top 25% rank scores were regarded as the main ones (Supplementary Fig. S8b).

Cluster 1 showed the highest rank score (score: 12.383) with 48 genes that included three TWAS genes, 44 meta-signature genes, and one gene from the STRING database (Fig. 3a). In cluster 1, *marker of proliferation Ki-67* (*MKI67*) was the hub gene with 30 degrees and 0.254 betweenness centrality (BC). Cluster 2 contained 30 upregulated and two downregulated meta-signature genes and 11.355 rank score (Fig. 3b). The hub gene for cluster 2 was *interferon regulatory factor 7* (*IRF7*) that presented 25 degrees and 0.272 BC. Cluster 3 had an 11.13 ranked score and consisted of the most nodes (116) with seven TWAS genes, 107 meta-signature genes, *FLG* (which was involved in both TWAS genes and meta-signatures), and one gene added by the STRING database. *MMP9*, which was an upregulated meta-signature, was the hub gene for cluster 3, showing 38 degrees and 0.192 BC (Fig. 3c). The connections between TWAS genes and meta-signatures in cluster 1 had the highest rank score and cluster 3 harbored the most genes. This suggests that the combination of TWAS genes and meta-signatures successfully expanded the genetic signatures of AD.

Additionally, we analyzed the connections between genes from our analyses and known AD-associated genes in functional networks specific to three tissues (blood, blood plasma, and skin) and 12 cell types (B-lymphocytes, culture condition CD8 cells, dendritic cells, eosinophils, granulocytes, keratinocytes, monocytes, mononuclear phagocytes, natural killer cells, neutrophils, skin fibroblasts, and T-lymphocytes)[29]. We compared the gene–gene functional connectivity of known AD markers and 289 genes from our analyses versus the connectivity of AD markers and randomly selected 289 genes. In all 15 networks, genes from our analyses showed significantly higher connectivity ($P < 0.001$, one-tailed Mann–Whitney) with known AD markers than random genes, suggesting their tissue- and cell-specific functional involvement in AD etiology (Supplementary Fig. S9).

**Identifying potential drug candidates for AD.** Using TWAS genes and meta-signatures, we discovered drug candidates for AD via a drug-repositioning approach. The connectivity map (CMAP) database contains the genome-wide transcriptional change data after the addition of small molecules (perturbagens). Enrichment scores of TWAS genes (TWAS-ES) and meta-signatures (Meta-ES) for each perturbagen were calculated using CMAP to select perturbagens with product scores >0.6 (Supplementary Data 6). Perturbagens selected as potential drug candidates were pararosaniline (TWAS-ES: 0.875; Meta-ES: 0.981; product score: 0.858), 2-deoxy-D-glucose (TWAS-ES: 0.916; Meta-ES: 0.936; product score: 0.857), cantharidin (TWAS-ES: 0.839; Meta-ES: 0.869; product score: 0.729), MG-132 (TWAS-ES: 0.683; Meta-ES: 0.984; product score: 0.672), and

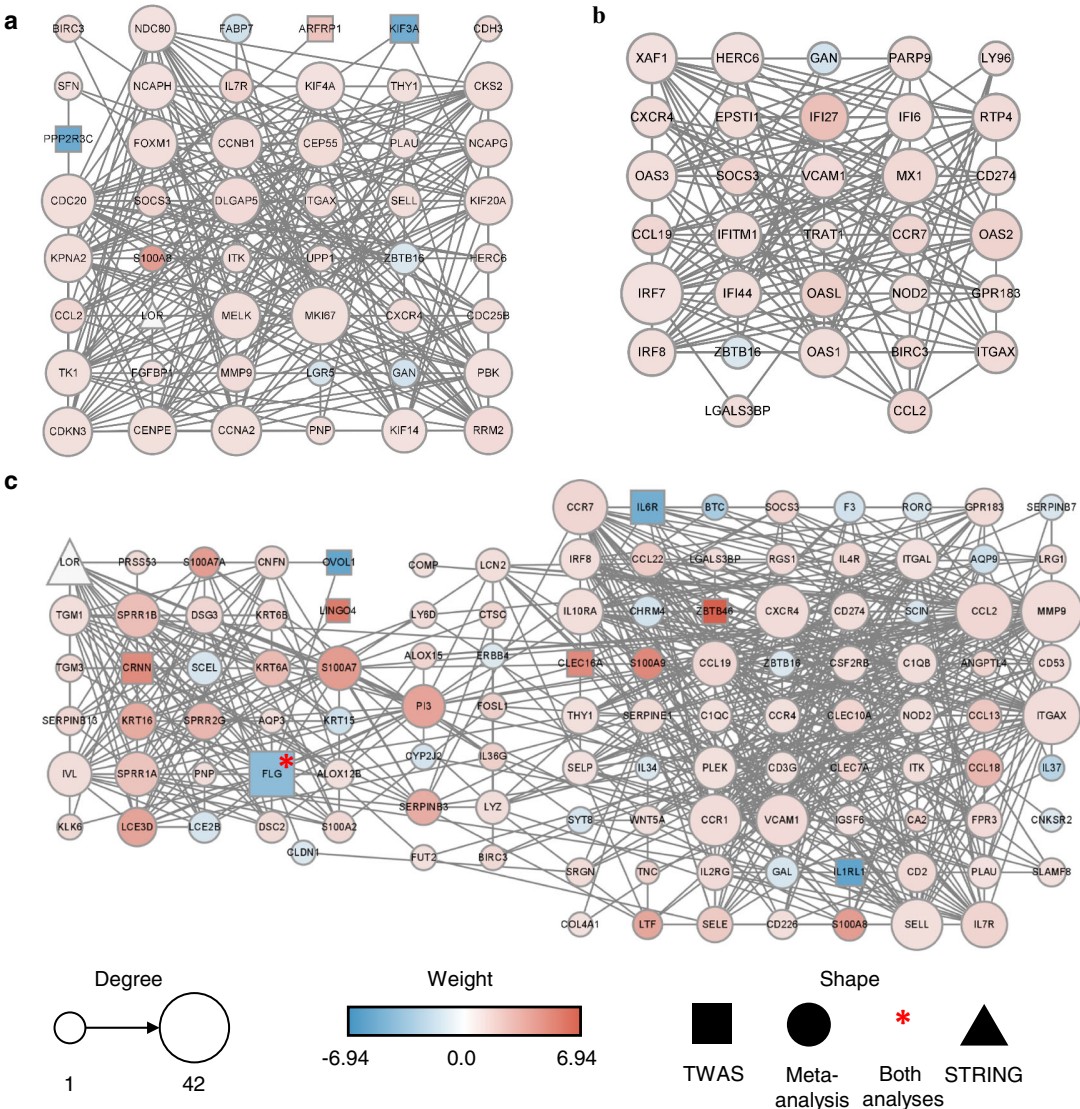

**Fig. 3 Sub-networks of the PPI network constructed with the functional protein association retrieved from the STRING database using the TWAS genes and meta-signatures.** The PPI network of the sub-network clustered using MCODE that were **a** cluster 1, **b** cluster 2, and **c** cluster 3. The size of each node is proportional to the degree of the node. The weight of each node (Z (TWAS) or meta-analysis log$_2$FC) is indicated by the color of the node. The shape of the node indicates where the gene came from. A circle, rectangle, or triangle corresponds to genes involved in TWAS, meta-analysis, and the STRING database, respectively. Significantly associated genes in both TWAS and meta-analysis are marked with a red asterisk.

1,4-chrysenequinone (TWAS-ES: 0.836; Meta-ES: 0.736; product score: 0.615) (Fig. 4a). To assess coherence between the drug lists derived from the two different sources, we analyzed the correlation between TWAS-ES and Meta-ES; those of each CMAP drug that were significantly enriched ($P < 0.01$) in both TWAS and meta-analysis were positively correlated ($R = 0.414$, $P = 2.791 \times 10^{-11}$), indicating that the significantly enriched drugs from TWAS and meta-analysis methods had significant coherence (Fig. 4b).

Finding structurally or functionally similar molecules to currently used drugs is a basic approach for drug repositioning. Therefore, we assessed the similarities of structures and modes of actions (MOAs) between our drug candidates and four reference drugs used to treat AD selected from three categories: tacrolimus as a topical calcineurin inhibitor, hydroxyzine and diphenhydramine as antihistamines, and cefalexin as an antibiotic[30–32]. We compared the chemical structures of our potential drug candidates and the reference drugs using the cosine coefficient (Fig. 4c). Our drug candidates showed a cosine coefficient in the

range 0.222–0.544 compared with reference drugs. Cantharidin and 2-deoxy-D-glucose were similar to the reference drug tacrolimus, and MG-132 to cefalexin and diphenhydramine, suggesting their high potential as treatment options for AD.

We carried out network-based MOA analysis to investigate the similarities in the transcriptional signatures of the drug candidates and reference drugs. Each drug candidate connected with at least one reference drug either directly or with just one stopover, as shown in Fig. 4d. 2-deoxy-D-glucose was directly connected to hydroxyzine and indirectly connected to tacrolimus, which showed structural similarity with tyrphostin as a stopover[33]. Pararosaniline had two indirect paths via a merged gene signature from PEGylated liposomal doxorubicin (PLD+) or an actin polymerase inhibitor, cytochalasin B, connected to hydroxyzine[34,35]. Both 1,4-chrysenequinone and cantharidin were directly connected to the reference drug cefalexin.

We identified potential drug candidates by analyzing gene lists from TWAS and transcriptome meta-analysis with CMAP that showed substantial similarities with currently used drugs in terms

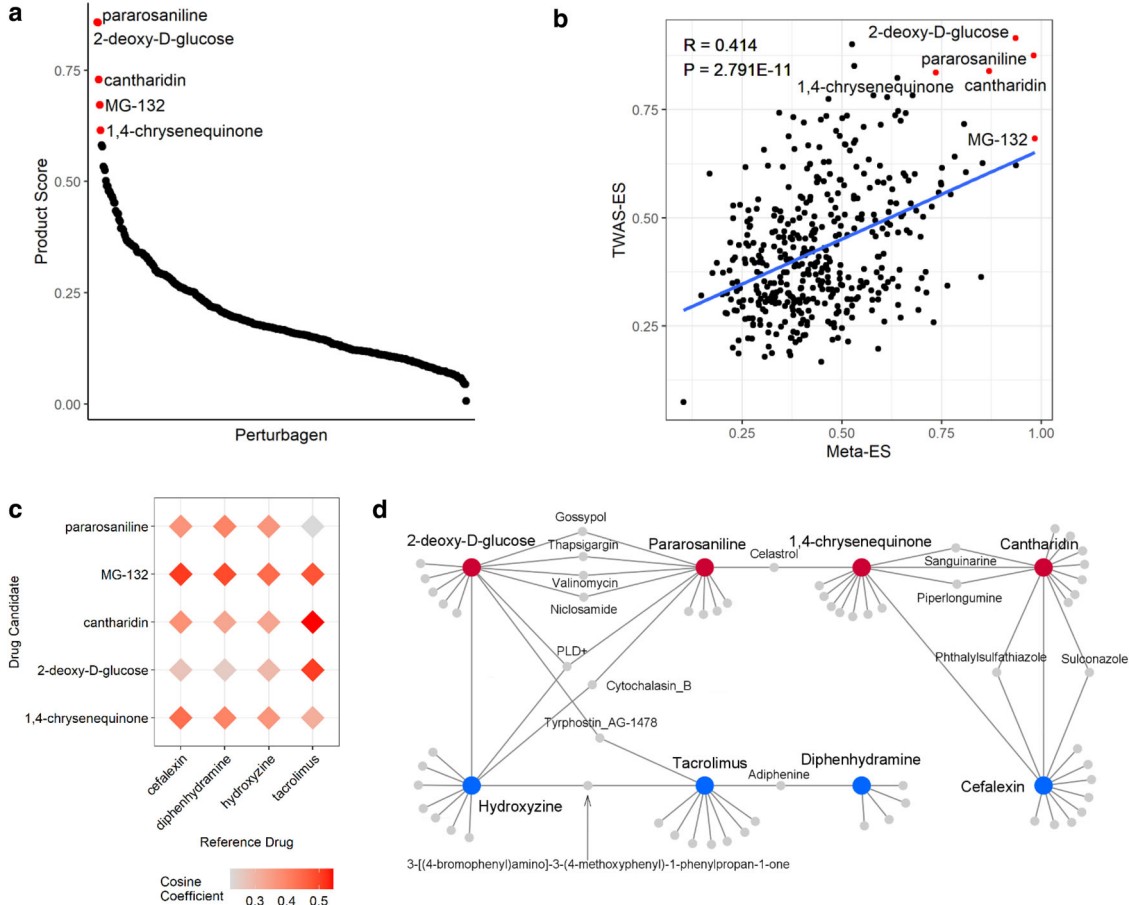

**Fig. 4 Identification of potential drugs for AD through in silico drug repositioning. a** A scatter plot of the calculated product score. Highly enriched drugs (product score > 0.6) are marked with red and annotated. **b** A scatter plot showing the correlation between the enrichment of perturbagens calculated with TWAS genes and meta-signatures. **c** The structure similarity analysis results comparing the potential drug candidates and reference drugs. The intensities of red rhombi are proportional to the cosine coefficient similarity index. **d** A network showing the similarities in MOAs of potential drug candidates and reference drugs. Red and blue nodes correspond to the potential drug candidates and reference drugs, respectively.

of chemical structures and MOAs, suggesting their potential for ameliorating AD symptoms.

## Discussion
TWAS calculates the expected gene expression values based on large-scale GWAS, of which the sample number usually exceeds those of transcriptome experiments from clinical studies. By predicting tissue-specific expression levels of AD using TWAS, we could identify four novel genes (Fig. 1a). *LINGO4* is a gene encoding a protein with an Ig-like C2 type domain and 13 leucine-rich domains. A previous study indicated the association between *LINGO4* and essential tremor in a Chinese population, but the contribution of *LINGO4* to AD has not been revealed, to the best of our knowledge[36]. The gene product of *RFX5* is reported to be associated with interferon gamma activation or major histocompatibility complex II gene expression, suggesting its role in AD pathogenesis[37–39]. Several studies mentioned the *P4HA2* gene in AD or AD-like symptoms, but none of these reports highlighted *P4HA2* as a major risk factor for AD[40–42]. The *RBM17* gene encodes a protein that induces cell cycle-related biological pathways[43]. This gene was mentioned in previous reports but was never highlighted as a main causal genetic risk factor for AD[44,45]. While recent research by Sobczyk et al. utilized the GWAS summary statistics from the EAGLE Consortium, which is the largest multi-ancestry study containing the genotypes of AD patients and healthy controls of European, African,

Japanese, and Latin American ancestry, we used the summary statistics of a European population from UK Biobank[45]. For this reason, we may have estimated genetic risk factors for AD in the European population more precisely, thereby identifying genes that were not found in the previous study.

Functional annotation of TWAS signals also conformed to known characteristics of AD pathogenesis (Fig. 1b). The most well-known genetic risk factor, *FLG*, is associated with the cornified envelope and peptide cross-linking, which are representative characteristics of AD and trigger skin barrier dysfunctions[46–48]. Enriched pathways in blood-related panels were related to immune responses such as the function and regulation of type 1 helper T cells, which are a signature of the transition from early- to chronic-stage AD[49].

Our meta-analytic approach combined five independent transcriptome datasets from previously published studies into a merged set with adjusted batch effects (Fig. 2a). Even though transcriptome meta-analyses have been previously performed, our study used 233 samples, which is the largest sample to date[50–52]. Because statistical power improves by increasing sample size, we obtained a meta-signature showing clear expression patterns across the samples and identified five novel genes, *C1orf162*, *NOCT*, *TIGAR*, *SCIN*, and *BOC*, that may play crucial roles in AD pathogenesis (Fig. 2b, c). Notably, TWAS signals were enriched in hedgehog signaling, and we identified the *BOC* gene, which plays a role in hedgehog signaling, from the meta-analysis (Figs. 1b and 2). The pathogenetic role of hedgehog

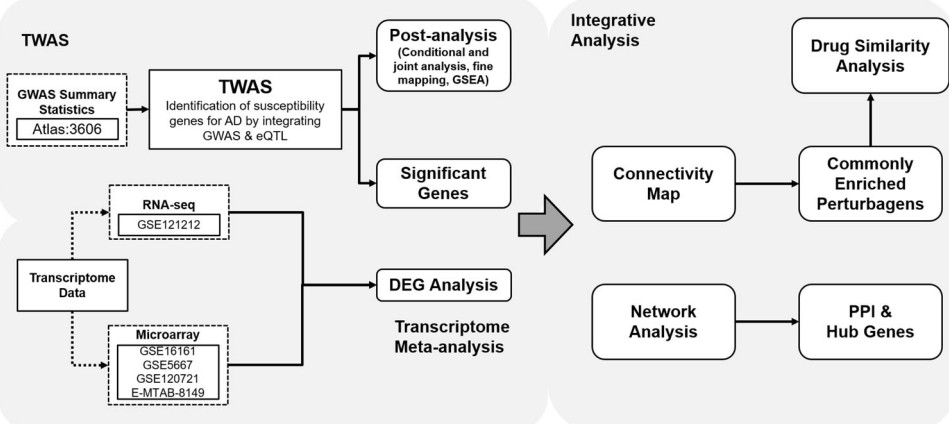

**Fig. 5 A schematic workflow of an integrative transcriptome-wide analysis for AD.** Left panel delineates the identification of candidate genes and right panel describes the integrative analysis for analyzing the gene-gene connectivity and identifying the drug candidate for AD.

signaling in AD has received some attention in recent experimental studies, and our study also revealed the connection between AD etiology and the abnormal activation of this signaling pathway.

TWAS has advantages in its sample size and statistical power for detecting genetic risk factors and their associated genes, whereas transcriptome studies measure expression values. We believe that integrating these two approaches complements what each method lacks. TWAS genes and meta-signature genes of AD were connected in two major sub-networks on the PPI network, suggesting that these gene connections may relate to AD pathogenesis (Fig. 3a–c).

We calculated product score using TWAS genes and meta-signature as inputs and identified five potential drugs for AD: pararosaniline, 2-deoxy-D-glucose, cantharidin, MG-132, and 1,4-chrysenequinone (Fig. 4a–d). Pararosaniline, 2-deoxy-D-glucose, cantharidin, and their derivatives had in vivo and/or clinical evidence of ameliorating various dermatological conditions[53,54]. MG-132 and 1,4-chrysenequinone inversed the gene expression patterns of AD in our in silico approach. Pararosaniline is an organic compound used as a fixation dye for frozen tissues or for the detection of aldehydes in biological materials[55]. Gentian violet, a hexamethyl form of pararosaniline, was previously used as an antibiotic, but has recently received attention for its potential to treat dermatologic diseases such as hypereosinophilic syndrome and pachyonychia congenita[52]. The glucose derivative 2-deoxy-D-glucose is used as an imaging agent for in vivo fluorescence imaging and has been implicated in targeted cancer therapies[56,57]. It also significantly ameliorates skin inflammation in dermatitis mouse models[53]. Cantharidin is a natural terpenoid compound produced in blister beetles, which were used in ancient Asia to treat conditions such as arthritis, pneumonia, ulcers, and smallpox[54]. Recent studies used cantharidin to manage dermatologic diseases like molluscum contagiosum and warts[58,59]. MG-132 is a proteasome inhibitor with anti-cancer activities that can also temporally alleviate AD-like symptoms in a murine model[60,61]. 1,4-chrysenequinone, a para-quinone antioxidant is associated with antigen presenting and processing[62–64]. Several studies have suggested 1,4-chrysenequinone as a therapeutic agent for cancerous diseases[65,66]. While our drug candidates showed moderate structural similarity with known AD drugs (0.222 < cosine coefficient < 0.544), we observed suggestive similarities in MOAs.

We combined two powerful approaches, TWAS and transcriptome meta-analysis, to investigate the complicated biological nature of AD and identified potential therapeutics through in silico drug repositioning (Fig. 5). We identified novel genetic factors associated with AD risk and/or pathogenesis, which have roles in skin barrier abnormality, immune cell dysregulation, cell cycles, and immune responses, through an integrative transcriptome approach. Because we used an in silico approach, our results may need to be validated with experimentation. While animal models for AD are available, they are imperfect representations of human AD and only have an AD-like phenotype[67,68]. Transcriptomic profiles of each murine model with AD-like phenotypes showed significant differences from human AD, indicating that our genetic markers need to be validated in human patients[69]. However, since our drug candidates are associated with ameliorating the symptoms of AD, the effectiveness could be validated using in vitro and in vivo models. We believe that our systematic large-scale analysis will expand the understanding of the biological phenomena underlying AD in humans.

## Methods

**Data collection and pre-processing for TWAS**. GWAS summary statistics for AD (Atlas ID: 3606; total: 289,307; control: 279,476; AD: 9831) based on UK Biobank (UKB2) were retrieved from GWAS Atlas (https://atlas.ctglab.nl/)[70,71]. The retrieved data were then converted into the LD score format using the LDSC software (version 1.0.1)[72]. An LD structure from the 1000 Genomes Project was used as the reference LD block for TWAS[73]. Seven eQTL panels from the GTEx project version 7 (skin-sun exposed, skin-not sun exposed, cells-transformed fibroblast, spleen, thyroid, whole blood and cells-EBV-transformed lymphocytes), and two eQTL panels from individual studies (NTR and YFS blood panel) were used as the pre-computed tissue-specific gene expression weights for TWAS[74–76]. The reference LD structure and eQTL panels were curated in the FUSION webpage (http://gusevlab.org/projects/fusion/) and used for TWAS of AD GWAS summary statistics[12].

**Transcriptome data collection and processing**. Transcriptome data were searched in Gene Expression Omnibus (GEO, https://www.ncbi.nlm.nih.gov/geo/) and ArrayExpress (https://www.ebi.ac.uk/arrayexpress/). Raw expression data and counts were retrieved for microarray datasets and RNA-seq data, respectively. Data derived from skin tissues of AD patients and healthy control groups were selected. The selected data consisted of one RNA-seq experiment (GSE121212, 38 controls and 27 AD patients) and four microarray experiments (GSE16161, GSE5667, GSE120721, E-MTAB-8149 with 9, 5, 22, and 19 controls and 9, 6, 15, and 83 AD patients, respectively). RNA-seq data were processed and normalized using edgeR R package, and the counts per million (cpm) were calculated with DESeq2 R package[77–79]. Microarray data were normalized using the robust multi-array average method in the oligo R package[80].

**Tissue-specific enrichment analysis of GWAS signals**. Tissue specificity analysis based on the GWAS data was conducted with the GENE2FUNC process of the FUMA web server[17]. The threshold for enrichment significance was Bonferroni-corrected $P < 0.05$. Tissue-specific heritability enrichment analysis was performed with LDSC-SEG on the multi-tissue expression and chromatin datasets that contained the tissue-specific gene expressions and epigenetic chromatin modifications, respectively[18]. Tissues with FDR < 0.05 were regarded as significantly enriched.

**Transcriptome-wide association analysis**. FUSION performs summary-based gene expression imputation to identify the association between expected gene expression values and the trait by applying weighted-linear mixed models using pre-computed eQTL panels composed of *cis*-effects on SNP-gene regulation and SNP-trait effects. TWAS for AD summary statistics was performed using the default parameters of FUSION. Gene expression was calculated with four models: best linear unbiased predictor, Bayesian sparse linear mixed model, elastic net, and least absolute shrinkage and selection operator. The result from the best performing model of each gene was displayed as the expected gene expression value. A permutation test was performed using FUSION to evaluate the robustness of the TWAS signals (number of permutations: 100,000).

**Gene prioritization analysis**. The MAGMA was performed with the FUMA web server (https://fuma.ctglab.nl/), and the COLOC analysis was implemented for the genes that showed $P < 0.05$ with FUSION software[17,19]. The significance threshold for the MAGMA was determined as a Bonferroni-corrected threshold ($P < 0.05$/the number of analyzed genes (18,899) = ~$2.64 \times 10^{-6}$). Each hypothesis represents the following phenomenon in our analysis. $H_0$: there is no causal variant; $H_1$: there are only causal variants between genotype and phenotype; $H_2$: there are only causal variants for eQTL; $H_3$: phenotype and gene expressions are driven by two different causal variants; and $H_4$: phenotype and gene expressions share the same causal variant. Following Li et al., we determined the threshold of colocalization as $PP3 + PP4 > 0.8$ and $PP4/PP3 > 2$[81].

**Post-analysis of TWAS results**. To assess the associations of multiple TWAS signals in the same loci, we conducted conditional and joint analysis for TWAS-significant loci with a FUSION post-process function. To support the robustness of novel TWAS signals, we performed fine-mapping of TWAS associations using the FOCUS method (version 0.6.10) proposed by Mancuso et al., while eQTL panels were confined to the tissue where TWAS-significant loci of interest were observed[82]. FOCUS identifies credible gene sets containing causal genes at the nominal confidence level (over 90%). Additionally, the biological pathways related to TWAS signals were analyzed by GSEA using a TWAS-GSEA (v.1.2, https://github.com/opain/TWAS-GSEA) with GO-BP and KEGG reference gene sets retrieved from the molecular signatures database (MsigDB, http://software.broadinstitute.org/gsea/msigdb)[21,22,83–85]. Tissue-specific effects of TWAS results were analyzed by calculating the mean of squared Z (TWAS) for each tissue following Mancuso et al.[86].

**Transcriptome meta-analysis**. Individual datasets were merged by corresponding the common Entrez IDs. The cpm values of the RNA-seq dataset were adjusted as $\log_2(cpm + 0.25)$ to avoid negative values following Mooney et al. with slight modifications[87]. Briefly, cpm values were used instead of fragments per kilobase per million mapped reads (FPKM) values. Batch effects between datasets were corrected using the ComBat function in the sva R package[88]. DEGs between the control group and AD group were identified using the limma R package[89]. DEGs with positive and negative $\log_2FCs$ were regarded as upregulated and downregulated meta-signatures, respectively.

**Validating correlation between TWAS results and meta-analysis**. GSEA was performed to examine the functional correlation between TWAS results from each panel and the results from transcriptome meta-analysis. GSEA pre-ranked method was performed on the gene sets with up- or downregulated meta-signatures and TWAS results ranked with the Z (TWAS) values from each panel. The significance threshold for enrichment was set as $FDR < 0.25$ following the recommendation of MsigDB. Functional annotation of the meta-analysis results was performed with GSEA pre-ranked method by ordering the genes by their $\log_2FC$ values. To analyze the overlapping enrichment with TWAS-results, we applied the gene sets used for TWAS-GSEA as the reference gene sets.

**Network analysis**. The significant genes from TWAS and DEGs from transcriptome meta-analysis were used as the input nodes for network analysis. STRING (https://string-db.org/) was used to construct PPI networks[90]. Constructed networks were processed using Cytoscape (version 3.8.2), and sub-network analysis was performed with the MCODE Cytoscape plug-in and the NetworkAnalyzer Cytoscape tool[91–93].

The list of the 2817 known AD-associated markers was downloaded from Open Targets Platform (https://platform.opentargets.org/)[94]. Tissue- or cell-specific functional networks were retrieved from HumanBase (https://hb.flatironinstitute.org/), and 15 AD-related tissue- or cell-specific networks were selected[29]. Selected networks were for three tissues (blood, blood plasma, and skin) and 12 cell types (B-lymphocytes, culture condition CD8 cells, dendritic cells, eosinophils, granulocytes, keratinocytes, monocytes, mononuclear phagocytes, natural killer cells, neutrophils, skin fibroblasts, and T-lymphocytes). Because the edge weights were extremely skewed and we did not want to select 'not-available' values, $\log_2(connectivity score + 1)$ was used to scale them. They were then analyzed with a one-tailed Mann–Whitney test.

**Drug repositioning with computational tools**. The CMAP is a web-based drug-repositioning tool that analyzes the input up- and down-gene signatures of in vitro-derived drug signatures in the CMAP database (https://portals.broadinstitute.org/cmap/) by Kolmogorov–Smirnov statistics[95]. TWAS-significant genes and meta-signatures were separately used as input for the analysis. Both gene lists were converted to the corresponding Affymetrix probe identifiers, and the queries were executed by reversing the AD signatures. Enrichment scores for each drug were combined by calculating individual product scores following Liu et al., and candidates with a product score $> 0.6$ were selected[96].

**Similarity analysis with currently approved drugs for AD**. The connectivity between approved AD drugs and our drug candidates was assessed following Kim et al.[97]. Among approved AD drugs, small molecules that are available in MANTRA 2.0 were selected as reference drugs. MOA similarities were analyzed with the MANTRA 2.0 web-based platform[98]. The maximum number of neighboring nodes was set to 10, and the MOA similarity network was visualized by Cytoscape (version 3.8.2). Structural information on the molecules in .sdf format was retrieved from DrugBank (https://drugbank.ca) and PubChem (https://pubchem.org) using the rcdk R package[99]. For comparison of structural similarities, the extended connectivity fingerprint with a diameter set to 4 was calculated for each molecule, and the cosine coefficients between the drug candidates and the reference drugs were calculated with the Rcpi R package[100].

**Statistical analysis**. Statistical analyses were conducted using the statistical computing programming language R (version 4.0.3). The results were visualized with R package ggplot2 and ggrepel (https://github.com/slowkow/ggrepel).

**Reporting summary**. Further information on research design is available in the Nature Research Reporting Summary linked to this article.

## Data availability

The GWAS summary statistics used in this study can be found in GWAS Atlas (https://atlas.ctglab.nl/) with the accession ID 3606. Multi-tissue expression or chromatin datasets for LDSC-SEG analysis can be found in following github page (https://github.com/bulik/ldsc/wiki/Cell-type-specific-analyses). Tissue-specific eQTL panels can be found in GTEx Portal (https://gtexportal.org/home/), and pre-computed weights can be downloaded from the FUSION web page (http://gusevlab.org/projects/fusion/). Transcriptome data from AD patients are available in NCBI-GEO (GSE121212, GSE16161, GSE5667, and GSE120721) and EBI-ArrayExpress (E-MTAB-8149). Previously reported AD marker genes were searched on Open Targets Platform (https://platform.opentargets.org/). Tissue- and cell type-specific reference networks were retrieved from HumanBase (https://hb.flatironinstitute.org/). Functional gene sets retrieved from MsigDB (http://software.broadinstitute.org/gsea/msigdb) were used in this study.

## Code availability

The following tools, software, and packages were used in this study: FUMA: https://fuma.ctglab.nl/; FUSION: http://gusevlab.org/projects/fusion/; LDSC, version 1.0.1: https://github.com/bulik/ldsc; FOCUS, version 0.6.10: https://github.com/bogdanlab/focus; TWAS-GSEA, version 1.2: https://github.com/opain/TWAS-GSEA; sva, version 3.34.0: https://www.bioconductor.org/packages/release/bioc/html/sva.html; limma, version 3.42.2: https://www.bioconductor.org/packages/release/bioc/html/limma.html; oligo, version 1.54.1: https://www.bioconductor.org/packages/release/bioc/html/oligo.html; edgeR, version 3.32.1: https://www.bioconductor.org/packages/release/bioc/html/edgeR.html; DESeq2, version 1.26.0: https://www.bioconductor.org/packages/release/bioc/html/DESeq2.html; GSEA, version 4.1.0: https://www.gsea-msigdb.org/gsea/index.jsp; STRING: https://string-db.org/; Cytoscape, version 3.8.2: https://cytoscape.org/; HumanBase, https://hb.flatironinstitute.org/; CMAP, https://portals.broadinstitute.org/cmap/; MANTRA 2.0: https://mantra.tigem.it/; Rcpi, version 1.26.0: https://www.bioconductor.org/packages/release/bioc/html/Rcpi.html; rcdk, version 3.5.0: https://cran.r-project.org/web/packages/rcdk/index.html; and ggrepel, version 0.8.2: https://cran.r-project.org/web/packages/ggrepel/index.html.

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

## Acknowledgements

This work was supported by the National Research Foundation of Korea (NRF) grant funded by the Korea government (MSIT) (No. NRF-2021R1A2C1008804). This research was supported by the MSIT (Ministry of Science and ICT), Korea, under the ITRC (Information Technology Research Center) support program (IITP-2022-2020-0-01789) supervised by the IITP (Institute for Information & Communications Technology Planning & Evaluation). This work was a part of the fulfillment of the requirements for Jaeseung Song's Ph.D. degree.

## Author contributions

J.S. and J.J. conceptualized and designed the study. J.S. collected data. J.S. analyzed data. J.S., D.K., and S.L. conducted formal analysis and data visualization. J.S., J.J., and W.J. interpreted data. J.S., J.J., and J.W.J.J. contributed to project administration. J.W.J.J. and W.J. supervised the research. J.S. wrote the original draft of the manuscript. J.W.J.J. and W.J. edited and revised the writing.

## Competing interests

The authors declare no competing interests.
