## [Peer Review File · Communications Biology]

Reviewers' comments:

Reviewer #1 (Remarks to the Author):

In order to identify genes associated with AD, the authors used the largest up-to-date AD GWAS dataset obtained from a European population to perform transcriptome meta-analysis with microarray and RNA sequencing (RNA-seq) datasets. Then, the authors performed in silico rug repositioning by combining the results from TWAS and meta-analysis to identify alternative therapeutic options to treat AD. This study has potentially provided novel candidate genes and drugs associated with AD, which may pose new insight for understanding mechanisms of AD. However, several concerns are required the authors to clarify.

(1). Through TWAS, this study has discovered many genes associated with AD, which include many novel findings. Currently, many approaches have been developed to link GWAS SNP index with genes. It is necessary to compare the TWAS results of this study with other methods to illustrate the novel findings of using the largest AD GWAS dataset in TWAS.

(2). The authors have shown many significant associations identified by this study. Are the significant findings enriched with known AD associated genes? Are the novel findings more likely to interact with known AD genes than random genes?

(3). No tissue-specific or cell-specific analysis was performed to indicate the novel genes highly enriched in AD related tissues or cells.

(4). This study constructed the gene-gene interaction network by STRING. If the author can construct the network according to gene-expression data from the AD-related tissues, the network is tissue-specific and is more accurate to illustrate the relationship between genes.

(5). Several candidate drugs were recommended by the authors. How many of the novel candidate drugs do have the similar drug structures as the known drugs?

Reviewer #2 (Remarks to the Author):

Here the authors seek to leverage published AD GWAS data and eQTL data across multiple tissue types, to perform TWAS analyses for AD. They also examine skin gene expression datasets in a meta-analysis to identify genes whose expression is associated with AD status to provide context for TWAS genes. The authors also perform other computational analyses on TWAS and meta-analysis genes, including network and drug repurposing analyses, to bring understanding/translational value to the genomic analysis findings. Overall, the analyses seem to be well-designed and performed, using appropriate methods and bioinformatic tools. However, I struggle to find any important novelty/value in the results without some sort of functional follow-up. The paper reads as a catalog of results with little context or implication provided. We don't even know the directionality of the effect for the novel genes, the tissues they are expressed in, or potential disease mechanisms. Easy opportunities to investigate and present tissue-specific effects are missed. In vitro, single cell, or other functional studies on the novel genes would have really increased my interest in the paper. I appreciate the drug repurposing analyses, but without follow-up in vitro analyses, it means little. In summary the lack of novelty or functional data makes this paper of little interest to the field. Other specific comments listed below.

1. The authors had an opportunity to compare TWAS signals between tissues to gain some sort of understanding regarding tissue specific genetic effects on AD. However, a thorough investigation of this is not completed, in fact the authors don't even state in the results section any interesting tissue-specific effects. This is a missed opportunity for some novelty.

2. The authors should state the tissues used in the TWAS analysis in the results section. Justification for using these tissues should be stated in the methods if not results.

3. The authors should state the GWAS data used in the results text as well as subject numbers.

4. I am surprised the combat-based batch correction was able to remove batch effects well. In the figure 2a PCA plot can the authors label samples by dataset to confirm batch effects are removed?

Reviewer #3 (Remarks to the Author):

Song et al. conducted TWAS using the large-scale AD GWAS summary statistics and tissue-specific gene expression weights from GTEx and other studies. They then performed transcriptome meta-analysis with microarray and RNA-seq datasets to identify differentially expressed genes. Subsequently, a network analysis was constructed using the results from TWAS and meta-analysis. Finally, they performed drug repositioning. This study is the first integrative analysis combining TWAS and meta-analysis for AD.

Major:

1. The authors used $P < 9.46 \times 10^{-7}$ and selected 25 TWAS genes associated with AD. Why 9.46×10^{-7} ? The developer of FUSION did not use this value as a filter cutoff in their publication (Gusev et al. 2016). Since the Bonferroni corrected P-value has been calculated, why not use the corrected adjusted P-value for filtering?
2. Except for the 5 novel genes mentioned in the article, are the other 263 meta-signatures already reported related to AD? If so, how do these genes perform in the results of TWAS?
3. How to explain the dramatic difference between the TWAS genes and meta-signatures? After all, there is only one overlap gene between two sets.
4. How about the functional enrichment of the 268 meta-signatures? Is it consistent with the results of TWAS-GSEA?
5. To what extent the results of drug repositioning can be validated? Can authors design or discuss the experiments to validate the identified potential drug candidates for AD?

Minor:

1. How many genes were used in the PCA of Figure 2A?
2. The P-value of Pearson's correlation coefficient is 0 or any value are very small? Can you give an accurate value?
3. Are there any small molecules in the CMAP database that have been used for AD treatment? What are their enrichment scores?
4. The author found 4 and 5 novel AD-related genes through TWAS and meta-analysis respectively. Compared with known genes, what new biological implications can these genes provide for our understanding of AD?
6. The x-axis of supplementary figure S2 is unclear.
7. What is the threshold for differentially expressed genes in transcriptome meta-analysis?

Response to Reviewer #1

First of all, the authors greatly appreciate the helpful comments that the reviewer made. The authors completely agree with the reviewer's commentaries and feel that all the comments were necessary. The authors believe that the issues raised by the reviewer were critical to improve the quality of the study, so the authors tried our best to intensively address the comments to the best of our knowledge. Additionally, due to the 5,000-words limitation, we rearranged or simplified some part of the descriptions in the results or methods section. To provide the detailed results or settings for softwares, we modified our supplementary figures and tables for additional results. Any modified sentences or phrases are highlighted in the revised version of the manuscript.

Comment 1. Through TWAS, this study has discovered many genes associated with AD, which include many novel findings. Currently, many approaches have been developed to link GWAS SNP index with genes. It is necessary to compare the TWAS results of this study with other methods to illustrate the novel findings of using the largest AD GWAS dataset in TWAS.

>>> **Response 1:** The authors totally agree with the reviewer's comment that the gene prioritization from GWAS results need to be compared with other methods. We believe this procedure may improve the reliability and rigorousness of our results. To address this issue, we additionally performed gene prioritization with two other methods: MAGMA and COLOC.^{1,2} The results of comparing 3 methods were added in the results section as below and the results were displayed as reviewer-only figure 1 and Supplementary Figure S4 (5p, line 7-19).

Then, we compared the TWAS results with two other gene prioritization methods: the multi-marker analysis of genomic annotation (MAGMA) and the COLOC method.^{1,2} While MAGMA analyzes the associated genes based on their chromosomal positions, COLOC is an R package for analyzing co-localization events to calculate posterior probabilities (PP) for hypotheses 0–4 (H_0 – H_4). We detected 68 genes significantly associated with AD using MAGMA by applying a Bonferroni-corrected threshold ($P < 2.64 \times 10^{-6}$) that overlapped with 12 genes from TWAS (Supplementary Figure S4A). The COLOC results showed 27 colocalized signals for AD ($PP_3 + PP_4 > 0.8$ and $PP_4/PP_3 > 2$), among which more than half (15/27) were also prioritized in TWAS (Supplementary Figure S4B and C). Among the 27 genes from COLOC, 13 overlapped with the results from MAGMA (Supplementary Figure S4C). Nine genes

were prioritized with all three methods: *OVOL1*, *ARFRP1*, *PPP2R3C*, *FAM177A1*, *CLEC16A*, *SLC2A4RG*, *ZBTB46*, *IL6R*, and *IL18RAP* (Supplementary Figure S4C).

Reviewer-only Figure 1. Comparison of TWAS with other gene-prioritization methods

(A) A Manhattan plot showing the position-based gene prioritization results conducted using the MAGMA. The red line indicates the Bonferroni significant threshold ($P < 2.64 \times 10^{-6}$) and the yellow dots correspond to the 68 significant genes. (B) A ternary plot showing the result of COLOC analysis. The color of the dots indicate whether the corresponding gene showed significant association in each method. Grey dots indicate the genes that were not significant in any of the methods. Blue and red dots correspond to the significantly associated

genes in COLOC and TWAS, respectively. The genes significantly associated in both COLOC and TWAS were marked with purple dots. (C) A Venn diagram comparing the significantly prioritized genes from TWAS, MAGMA, and COLOC.

>>> The descriptions and details for both methods were described in the method section (16p, line 15-24) as below.

Gene prioritization analysis

The MAGMA was performed with the FUMA web server (<https://fuma.ctglab.nl/>), and the COLOC analysis was implemented for the genes that showed $P < 0.05$ with FUSION software.^{1,3} The significance threshold for the MAGMA was determined as a Bonferroni-corrected threshold ($P < 0.05/\text{the number of analyzed genes (18,899)} = \sim 2.64 \times 10^{-6}$). Each hypothesis represents the following phenomenon in our analysis. H_0 : there is no causal variant; H_1 : there are only causal variants between genotype and phenotype; H_2 : there are only causal variants for eQTL; H_3 : phenotype and gene expressions are driven by two different causal variants; and H_4 : phenotype and gene expressions share the same causal variant. Following Li *et al.*, we determined the threshold of co-localization as $PP3+PP4 > 0.8$ and $PP4/PP3 > 2$.⁴

Comment 2. The authors have shown many significant associations identified by this study. Are the significant findings enriched with known AD associated genes? Are the novel findings more likely to interact with known AD genes than random genes?

>>> **Response 2:** The authors also agree with the reviewer's comment that clarifying the connections between the genes from our analyses and known AD-associated genes may

strengthen our results. Together with the comment 3 and 4, which required additional analyses in the tissue- or cell type-specific level, we validated the functional connection between our TWAS genes and meta-signatures with known AD markers by analyzing the tissue- and cell-specific networks (Reviewer-only Figure 2). The results were added in the results section as below (9p, line 11-20).

Additionally, we analyzed the connections between genes from our analyses and known AD-associated genes in functional networks specific to three tissues (blood, blood plasma, and skin) and 12 cell types (B-lymphocytes, culture condition CD8 cells, dendritic cells, eosinophils, granulocytes, keratinocytes, monocytes, mononuclear phagocytes, natural killer cells, neutrophils, skin fibroblasts, and T-lymphocytes).⁵ We compared the gene–gene functional connectivity of known AD markers and 289 genes from our analyses versus the connectivity of AD markers and randomly selected 289 genes. In all 15 networks, genes from our analyses showed significantly higher connectivity ($P < 0.001$, one-tailed Mann–Whitney) with known AD markers than random genes, suggesting their tissue- and cell-specific functional involvement in AD etiology (Supplementary Figure S9).

Reviewer-only Figure 2. The results of tissue- or cell-specific gene-gene connectivity analysis.

The box plots of the tissue- or cell-specific analysis results. Gene-gene connectivity score between randomly selected genes and known AD markers were compared with the connectivity score between genes from TWAS and/or meta-analysis and known AD genes. Red and blue boxes indicate the rescaled connectivity scores from the randomly selected genes and TWAS genes and/or meta-signatures, respectively. *** $P < 0.001$

>>> Also, the methods and data used for the analysis were listed in the methods section as below (18p, line 16-25).

The list of the 2,817 known AD-associated markers was downloaded from Open Targets Platform (<https://platform.opentargets.org/>).⁶ Tissue- or cell-specific functional networks were retrieved from HumanBase (<https://hb.flatironinstitute.org/>), and 15

AD-related tissue- or cell-specific networks were selected.⁵ Selected networks were for three tissues (blood, blood plasma, and skin) and 12 cell types (B-lymphocytes, culture condition CD8 cells, dendritic cells, eosinophils, granulocytes, keratinocytes, monocytes, mononuclear phagocytes, natural killer cells, neutrophils, skin fibroblasts, and T-lymphocytes). Because the edge weights were extremely skewed and we did not want to select ‘not-available’ values, $\log_2(\text{connectivity score}+1)$ was used to scale them. They were then analyzed with a one-tailed Mann–Whitney test.

Comment 3. No tissue-specific or cell-specific analysis was performed to indicate the novel genes highly enriched in AD related tissues or cells.

>>> **Response 3:** The authors are truly thankful for the reviewer’s comment that tissue- or cell-specific analysis may clarify our TWAS results and further provide abundant information to potential readers. Other than abovementioned network analysis, we additionally performed tissue specificity analysis for the GWAS data with GENE2FUNC process in FUMA web server and LDSC-SEG tool. In the GENE2FUNC process, we used GTEx v7 eQTL panels for 53 tissues to identify in which the genome-wide significant signals were enriched (Reviewer-only Figure 3).³ Additionally, to determine the tissue-specific heritability enrichments of the polygenic signals, LDSC-SEG was performed with the multi-tissue dataset that was used in the previous study by Finucane *et al.* (Reviewer-only Figure 4A and B).⁷ In addition, we calculated the mean chi-squared values of TWAS Z-scores for each tissue to examine gene expression of which tissue was mostly affected by the genomic variants (Reviewer-only figure 5).⁸ Following descriptions were added in the results section of the modified manuscript.

>>> The results of the analyses were briefly described in the result section of the revised manuscript as below. (3p, line 16 – 4p, line 3)

Enrichment analysis of GWAS signals from AD GWAS summary statistics

To examine the genetic landscape of AD, this study uses the UK Biobank GWAS data consisting of 279,476 controls and 9,831 AD patients. First, we examined whether the GWAS signals for AD were specifically enriched in certain tissue or cell types by using the functional mapping and annotation of genetic association (FUMA). We found that the *cis*-regulated genes of GWAS signals were mainly over-expressed in skin tissues (Supplementary Figure S1).³ Next, tissue- or cell-specific heritability was analyzed using a linkage disequilibrium (LD) score regression applied to specifically expressed genes (LDSC-SEG) using the multi-tissue expression dataset and multi-tissue chromatin dataset following Finucane *et al.*⁷ Heritability of AD GWAS signals on the multi-tissue expression data showed significant enrichment (false discovery rate (FDR) < 0.05) in the blood and immune-related tissues (Supplementary Figure S2A; Supplementary Table S1) and this pattern was replicated in the multi-tissue chromatin dataset (Supplementary Figure S2B; Supplementary Table S1).

Reviewer-only Figure 3. Tissue-specific gene enrichment analysis of GWAS signals using FUMA

The bar plots of the tissue-specific gene enrichment analysis results using AD GWAS summary data using the GENE2FUNC process in FUMA web server. Upper, middle, and lower panel shows the enrichment of up-, down-regulated or both sides of DEGs in tissues denoted in x-axis compared with other tissues, respectively. The height of the bars corresponds to the $-\log_{10} (P\text{-value})$ of the tissue type enrichment. Red bars indicate the significant enrichment of the tissue-specific expressions (Bonferroni corrected $P < 0.05$).

Reviewer-only Figure 4. Tissue or cell-specific heritability enrichment analysis of AD using LDSC-SEG

(A) A scatter plot showing the result of heritability enrichment analysis of AD GWAS summary statistics using the multi-tissue expression dataset. (B) A scatter plot of heritability enrichment results using the chromatin interaction data from multiple tissues or cell types. The color of the dots indicates the annotated category of the tissue or cell type as listed below each figure. For significantly enriched signals, the size of the dots are

proportional to the scaled coefficient values. Grey line indicates the significance threshold of heritability enrichment (FDR < 0.05). Full table of the enrichment profiles is provided as Supplementary Table S1.

>>> The results for tissue-specific effect analysis using TWAS were added as below. (4p, line 14-17)

Although TWAS signals showed the highest mean effect size in the skin-not sun exposed panel, this was not dramatically higher than the mean effect sizes of other panels, indicating that the genetic features of AD may evenly affect the gene expression levels of nine tissue panels (Supplementary Figure S3).

Reviewer-only Figure 5. Tissue-specific effect sizes of the TWAS signals

A bar plot showing the tissue-specific effects of TWAS signals. The x-axis and the shade of green of the bar indicate the mean of squared TWAS-Z scores that represents the effect size from overall TWAS signals in the corresponding tissue panel.

>>> Methods for analyzing tissue-specific effects of GWAS signals were described as follows. (15p, line 20 – 16p, line 2)

Tissue-specific enrichment analysis of GWAS signals

Tissue specificity analysis based on the GWAS data was conducted with the GENE2FUNC process of the FUMA web server.³ The threshold for enrichment significance was Bonferroni-corrected $P < 0.05$. Tissue-specific heritability enrichment analysis was performed with LDSC-SEG on the multi-tissue expression and chromatin datasets that contained the tissue-specific gene expressions and epigenetic chromatin modifications, respectively.⁷ Tissues with $FDR < 0.05$ were regarded as significantly enriched.

>>> Tissue-specific effect analysis was conducted as below and attached in the methods section of the revised manuscript. (17p, line 11-13)

Tissue-specific effects of TWAS results were analyzed by calculating the mean of squared TWAS Z-scores for each tissue as described in Mancuso *et al.*⁸

Comment 4. This study constructed the gene-gene interaction network by STRING. If the author can construct the network according to gene-expression data from the AD-related tissues, the network is tissue-specific and is more accurate to illustrate the relationship between genes.

>>> **Response 4:** The authors appreciate the helpful comment that our TWAS and meta-analysis results need to be analyzed in tissue- or cell-specific manner. As we described above, we analyzed the connectivity between the genes from our analyses and known AD markers. In all of the 15 tissue- or cell type-specific networks, genes from TWAS and meta-analysis showed significantly higher connectivity compared with randomly selected genes. We hope that our response could address this issue with satisfaction.

Comment 5. Several candidate drugs were recommended by the authors. How many of the novel candidate drugs do have the similar drug structures as the known drugs?

>>> **Response 5:** The authors feel very grateful for your helpful comment. We totally agree that directly mentioning the number of novel candidate drugs that showed the high structural similarity may bring the powerful novelty to our results. Regardless of the similarity calculation methods, the coefficient threshold of coefficient value > 0.85 is widely used as the cutoff value in the majority of cases; however, it was hard to determine whether this threshold is the gold standard value for determining the similarity of two molecules. By applying the conventional threshold of similarity, our potential drug candidates showed only moderate similarity (between 0.3 and 0.6) with reference drugs. Therefore, we modified the descriptions about our results in the discussion section as below by mentioning the structural similarity levels of drug candidates and their suggestive similarities in MOAs (14p, line 1-3)

While our drug candidates showed moderate structural similarity with known AD drugs ($0.222 < \text{cosine coefficient} < 0.544$), we observed suggestive similarities in MOAs.

Reference

- 1 Giambartolomei, C. *et al.* Bayesian test for colocalisation between pairs of genetic association studies using summary statistics. *PLoS Genet* **10**, e1004383, doi:10.1371/journal.pgen.1004383 (2014).
- 2 de Leeuw, C. A., Mooij, J. M., Heskes, T. & Posthuma, D. MAGMA: generalized gene-set analysis of GWAS data. *PLoS Comput Biol* **11**, e1004219, doi:10.1371/journal.pcbi.1004219 (2015).
- 3 Watanabe, K., Taskesen, E., van Bochoven, A. & Posthuma, D. Functional mapping and annotation of genetic associations with FUMA. *Nat Commun* **8**, 1826, doi:10.1038/s41467-017-01261-5 (2017).
- 4 Li, Y. I., Wong, G., Humphrey, J. & Raj, T. Prioritizing Parkinson's disease genes using population-scale transcriptomic data. *Nat Commun* **10**, 994, doi:10.1038/s41467-019-08912-9 (2019).
- 5 Greene, C. S. *et al.* Understanding multicellular function and disease with human tissue-specific networks. *Nat Genet* **47**, 569-576, doi:10.1038/ng.3259 (2015).
- 6 Ochoa, D. *et al.* Open Targets Platform: supporting systematic drug-target identification and prioritisation. *Nucleic Acids Res* **49**, D1302-D1310, doi:10.1093/nar/gkaa1027 (2021).
- 7 Finucane, H. K. *et al.* Heritability enrichment of specifically expressed genes identifies disease-relevant tissues and cell types. *Nat Genet* **50**, 621-629, doi:10.1038/s41588-018-0081-4 (2018).
- 8 Mancuso, N. *et al.* Large-scale transcriptome-wide association study identifies new prostate cancer risk regions. *Nat Commun* **9**, 4079, doi:10.1038/s41467-018-06302-1 (2018).

Response to Reviewer #2

First of all, the authors are grateful for the helpful comments that the reviewer made. The authors completely agree with the reviewer's comments and feel that all of the comments were necessary. The authors believe that the issues raised by the reviewer were critical to enhance the quality of the study, hence the authors tried our best to intensively address the comments to the best of our knowledge. Unfortunately, due to the complicated human AD pathogenetic mechanisms, animal models can only partially mimic AD at the phenotype level only but not at the genotype level. Cell line models cannot completely mimic the complete nature of AD. We believe that the only way to provide functional experiments fitting the concept will be human study, in which case warrants a whole new manuscript of its own. Thus, it was hard to carry out appropriate functional experiments fitting for our study concept. Instead, we performed additional *in silico* analyses including heritability enrichment analysis or tissue-specific effect analysis to improve our study in both quantity and quality. Additionally, due to the limits up to 5,000 words, we rearranged or simplified part of the descriptions in the results or methods section. To provide the detailed results or settings for softwares, we modified our supplementary figures and tables for additional results. Any modified sentences or phrases are highlighted in the revised version of the manuscript.

Comment 1. The authors had an opportunity to compare TWAS signals between tissues to gain some sort of understanding regarding tissue specific genetic effects on AD. However, a thorough investigation of this is not completed, in fact the authors don't even state in the results section any interesting tissue-specific effects. This is a missed opportunity for some novelty.

>>> Response 1: The authors totally agree with the reviewer's comment that a deeper discussion on our TWAS results in a tissue-specific manner will definitely improve the novelty of the study. To address this issue, we first briefly discussed the number of significant TWAS associations in each tissue. We then analyzed the mean squared TWAS Z-scores for each tissue in order to observe whether the expected transcriptional changes were enriched in specific tissue type (Reviewer-only figure 1). We modified the result section by mentioning the number of significant TWAS associations for each tissue (4p. line 17-21).

The numbers of significant associations were six in skin-sun exposed, five in skin-not sun exposed, five in cells-transformed fibroblast, one in spleen, seven in thyroid, eight in whole blood, one in cells-EBV-transformed lymphocytes, two in NTR blood, and three in YFS blood panel. These results may represent the tissue-specific genetic features of AD in skin functions, immunological abnormalities, and thyroid autoimmunity.

>>> The result of the tissue-specific enrichment analysis of TWAS signals was briefly described in the result section of the revised manuscript as below (4p. line 14-17).

Although TWAS signals showed the highest mean effect size in the skin-not sun exposed panel, this was not dramatically higher than the mean effect sizes of other panels, indicating that the genetic features of AD may evenly affect the gene expression levels of nine tissue panels (Supplementary Figure S3).

Reviewer-only Figure 1. Tissue-specific effect size of the TWAS signals

A bar plot showing the tissue-specific effects of TWAS signals. The x-axis and the shade of green of the bar indicate the mean of squared TWAS-Z scores that represents the effect size from overall TWAS signals in the corresponding tissue panel.

>>> Also, we attached the following sentence in the materials and methods section of the revised manuscript (17p, line 11-13).

Tissue-specific effects of TWAS results were analyzed by calculating the mean of squared TWAS Z-scores for each tissue as described in Mancuso *et al.*¹

Comment 2. The authors should state the tissues used in the TWAS analysis in the results section. Justification for using these tissues should be stated in the methods if not results.

>>> Response 2: The authors are truly thankful and also agree with the reviewer's comment that clarifying the tissue panels used for TWAS need to be listed in the beginning of the results section. We modified the following description in the results section as below (4p. line 6-11).

To identify susceptibility genes for AD, we performed TWAS with functional summary-based imputation (FUSION), using eQTL panels from nine tissues that can cover the systemic features of AD. The tissue panels were skin-sun exposed, skin-not sun exposed, cells-transformed fibroblast, spleen, thyroid, whole blood, cells-Epstein–Barr virus (EBV)-transformed lymphocytes, Netherlands Twin Registry (NTR) blood, and Young Finns Study (YFS) blood panel.

Comment 3. The authors should state the GWAS data used in the results text as well as subject numbers.

>>> Response 3: We agree with the reviewer's comment that stating the GWAS data in the results section is necessary. We added the descriptions of GWAS data in the results section as below (3p, line 17-18)

To examine the genetic landscape of AD, this study uses the UK Biobank GWAS data consisting of 279,476 controls and 9,831 AD patients.

Comment 4. I am surprised the combat-based batch correction was able to remove batch effects well. In the figure 2a PCA plot can the authors label samples by dataset to confirm batch effects are removed?

>>> **Response 4:** The authors agree that presenting or PCA plot labeled by the dataset will better display the batch effect removal. We changed the shape of the dots by the dataset and we found that the batch effects by the dataset were also evenly adjusted (Reviewer-only Figure 2). We also decided to replace the original PCA plot with a newly created plot. Again, the authors are truly appreciated for the helpful comments.

Reviewer-only Figure 2. A modified PCA plot showing the differences between datasets.

A scatter plot displaying the PCA results using all genes after the batch effect correction. The shape of the points indicate the samples from each dataset. Green and red color correspond to the healthy control samples and AD samples, respectively.

Reference

- 1 Mancuso, N. *et al.* Large-scale transcriptome-wide association study identifies new prostate cancer risk regions. *Nat Commun* **9**, 4079, doi:10.1038/s41467-018-06302-1 (2018).

Response to Reviewer #3

First of all, the authors are grateful for all the comments that the reviewer made. The authors completely agree with the reviewer's comments and feel that all of the comments were necessary. The authors believe that the issues raised by the reviewer were critical to enhance the novelty of the study, so the authors tried our best to intensively address the comments to the best of our knowledge. Additionally, due to the limitations of 5,000 words, we had to rearrange or simplify some part of the descriptions in the results or methods section. To provide the detailed results or settings for softwares, we modified our supplementary figures and tables for additional results. Any modified sentences or phrases are highlighted in the revised version of the manuscript.

Major comments

Comment 1. The authors used $P < 9.46 \times 10^{-7}$ and selected 25 TWAS genes associated with AD. Why 9.46×10^{-7} ? The developer of FUSION did not use this value as a filter cutoff in their publication (Gusev et al. 2016). Since the Bonferroni corrected P-value has been calculated, why not use the corrected adjusted P-value for filtering?

>>> **Response 1:** The authors agree with the reviewer's comment that the statement of the P-value threshold for TWAS may confuse the readers. We did originally refer to the publication by Gusev et al., but it seems that our intention was not clearly stated. The significance threshold of $P < 9.46 \times 10^{-7}$ was indeed determined by using a Bonferroni-corrected threshold value. To prevent confusion, we changed all of the expressions "Bonferroni corrected P-value" into the "Bonferroni-corrected threshold". The authors greatly appreciate the reviewer for the helpful comment that can clarify our study.

Comment 2. Except for the 5 novel genes mentioned in the article, are the other 263 meta-signatures already reported related to AD? If so, how do these genes perform in the results of TWAS?

>>> **Response 2:** We selected our novel genes by searching the popular databases such as OpenTargetsPlatform (<https://platform.opentargets.org/>) and MalaCards (<https://malacards.org>).^{1,2} The genes that were not listed as the causal gene or marker gene of AD in both databases were suggested as the novel genes in our study. Unfortunately, there were no more TWAS-significant genes among meta-signatures. Except for *FLG* that was significant in both TWAS and meta-analysis, the gene with the lowest P (TWAS) was *involucrin (IVL)* with 1.58×10^{-5} and 4.32 of Z (TWAS). There were 4 genes in the range of $1 \times 10^{-5} < P \text{ (TWAS)} < 1 \times 10^{-4}$, one gene in $1 \times 10^{-4} < P \text{ (TWAS)} < 1 \times 10^{-3}$, and 8 genes in $1 \times 10^{-3} < P \text{ (TWAS)} < 1 \times 10^{-2}$. If there were multiple association results for a single gene, association with minimum P (TWAS) was counted. Detailed information is provided below as the Reviewer-only Table 1.

Reviewer-only Table 1. TWAS results of 263 meta-signatures. The associations with P (TWAS) $< 1 \times 10^{-2}$ were displayed.

Symbol	Entrez ID	Panel (TWAS)	P (TWAS)	Z (TWAS)	log ₂ FC (Meta-analysis)	FDR (Meta-analysis)
AQP3	360	Skin-sun exposed	7.23×10^{-5}	3.96	1.09	1.00×10^{-59}
		Skin-not sun exposed	8.89×10^{-3}	2.62		
CCR7	1236	Cells-Transformed fibroblasts	2.62×10^{-3}	3.01	1.50	5.56×10^{-75}
		Cells-EBV-transformed lymphocytes	8.72×10^{-5}	3.92		
CDHR1	92211	Skin-sun exposed	7.73×10^{-3}	2.66	-1.00	2.13×10^{-34}
CPXMI	56265	Thyroid	9.20×10^{-3}	2.60	1.03	4.10×10^{-27}
FAM189A2	9413	Thyroid	4.60×10^{-3}	2.83	-1.08	7.02×10^{-70}
FLG	2312	Cells-Transformed	4.27×10^{-10}	-6.24	-1.25	7.66×10^{-29}

		fibroblasts				
		Skin-sun exposed	8.77×10^{-14}	-7.46		
		Thyroid	8.67×10^{-13}	-7.15		
IL4R	3566	YFS Blood	3.08×10^{-3}	-2.96	1.34	1.58×10^{-103}
IVL	3713	Skin-not sun exposed	1.58×10^{-5}	4.32	1.19	2.72×10^{-33}
KPNA2	3838	YFS Blood	3.59×10^{-3}	-2.91	1.10	7.19×10^{-89}
PDZK1IP1	10158	NTR Blood	2.71×10^{-3}	-3.00	1.20	5.89×10^{-58}
		YFS Blood	7.02×10^{-3}	-2.70		
		Whole Blood	5.29×10^{-3}	-2.78		
S100A7	6278	Skin-sun exposed	7.28×10^{-3}	-2.68	4.20	7.06×10^{-89}
SNX10	29887	Skin-sun exposed	4.06×10^{-3}	-2.87	1.17	3.89×10^{-62}
SPRR18	6699	Skin-not sun exposed	9.03×10^{-4}	3.32	2.59	1.48×10^{-54}
		Skin-sun exposed	2.85×10^{-5}	4.18		

Comment 3. How to explain the dramatic difference between the TWAS genes and meta-signatures? After all, there is only one overlap gene between two sets.

>>> **Response 3:** The authors appreciate the helpful comment that the reviewer made and we totally agree that additional explanation for the difference between TWAS and meta-analysis is necessary. As we addressed in the comment #2, we were unable to find any additional significant relationship between TWAS genes and meta-signatures by comparing the genes directly. Instead, we performed additional gene set enrichment analysis using the gene sets (.gmt file) generated with the DEGs from meta-analysis and the pre-ranked gene list (.rnk file) ordered with the Z (TWAS) from each tissue panel. We found that the up-regulated gene set of skin-not sun exposed (suprapubic) and down-regulated gene set of skin-sun exposed (lower leg) panels were significantly correlated with the meta-signatures (Supplementary Figure S7, Reviewer-only Figure 1A and B). The results for the analysis were added in the results section of the revised manuscript as below. (8p, line 5-7)

Although there was only one direct overlap between TWAS genes and meta-signatures, we observed significant correlations between the two in gene-set levels (Supplementary Figure S7).

A**B**
Reviewer-only Figure 1. Gene set level correlation analysis of TWAS and meta-analysis using GSEA

(A) A GSEA enrichment plot for TWAS signals from skin – not sun exposed panel compared with up-regulated meta-signatures. (B) A GSEA enrichment plot comparing TWAS signals from skin – sun exposed panel and down-regulated meta-signatures.

>>> Also, methods for the analysis were described in the methods section as following. (17p, line 25 – 18p, line 8)

Validating correlation between TWAS results and meta-analysis

GSEA was performed to examine the functional correlation between TWAS results from each panel and the results from transcriptome meta-analysis. GSEA pre-ranked method was performed on the gene sets with up- or down-regulated meta-signatures and TWAS results ranked with the Z (TWAS) values from each panel. The significance threshold for enrichment was set as $FDR < 0.25$ following the recommendation of MsigDB. Functional annotation of the meta-analysis results was performed with GSEA pre-ranked method by ordering the genes by their \log_2FC

values. To analyze the overlapping enrichment with TWAS-results, we applied the gene sets used for TWAS-GSEA as the reference gene sets.

Comment 4. How about the functional enrichment of the 268 meta-signatures? Is it consistent with the results of TWAS-GSEA?

>>> **Response 4:** The authors also agree with the reviewer's comment that the comparison between the gene set level results of TWAS and meta-analysis may improve our study. To address this issue, we performed GSEA with the ranked gene list of meta-analysis results by weighting each gene with their \log_2 fold-change values. The gene sets used in TWAS-GSEA were identically tested for the GSEA of meta-analysis. In line with the GSEA comparing TWAS and meta-analysis directly, we detected that 80% (12/15) of significantly enriched gene sets from TWAS-GSEA showed significant enrichment with the results from meta-analysis. Again, we were able to check the gene set level consistency between TWAS and meta-analysis. Entire results of GSEA with meta-analysis results was newly added in Supplementary Table S5. The results of the analysis were described in the results section of the revised manuscript as below. (8p line 7-11)

In line with the significant enrichment of TWAS results in meta-signatures, the functional enrichment results of the meta-analysis well conformed with the TWAS-GSEA results. We found that 80% of gene sets that were significantly enriched with TWAS signals were also enriched with the pre-ranked gene lists generated using the transcriptome meta-analysis (Supplementary Table S5).

>>> Detailed methods for the analysis were added in the methods section as below. (18p, line 5-8)

Functional annotation of the meta-analysis results was performed with GSEA pre-ranked method by ordering the genes by their log₂FC values. To analyze the overlapping enrichment with TWAS-results, we applied the gene sets used for TWAS-GSEA as the reference gene sets.

Comment 5. To what extent the results of drug repositioning can be validated? Can authors design or discuss the experiments to validate the identified potential drug candidates for AD?

>>> **Response 5:** The authors also agree that extra validation works for the potential drug candidates are necessary to ensure the effect and safety of the molecules that we suggested in our manuscript. Because validating the efficacy of the drug candidates warrants an independent study of its own, we performed additional virtual docking using the TWAS-genes and meta-signatures against the potential drugs we suggested.

Additional virtual screening procedure was carried out with AutoDock Vina (ver. 1.2.0.) following previous protocols (ref, ref).^{3,4} Because some of the protein structures for TWAS-genes and meta-signatures had only been identified partially, protein structure files for 281 available genes in .pdb format were retrieved from AlphaFold protein structure database (<https://alphafold.ebi.ac.uk/>).^{5,6} Chemical structure data for our drug candidates in .sdf format were downloaded from DrugBank (<https://go.drugbank.com/>) or PubChem (<https://pubchem.ncbi.nlm.nih.gov/>). Structure data for both proteins and drugs were converted into .pdbqt format using ADFRsuite (version 1.0). Binding grids for each protein structure were generated using AutoGrid4. Virtual docking was performed with AutoDock

Vina with a basic docking method, applying default parameters for the scoring function from AutoDock4.

Usually recommended cut-off value for affinity output from AutoDock software is -14 kcal/mol; however, free energy \sim -7 to -9 kcal/mol is frequently regarded as suggestive value.^{7,8} In this point of view, 4 drug candidates (1,4-chrysenquinone, cantharidin, MG-132, and pararosaniline) met minimum suggestive binding energy with the protein structure of 281 genes, while 2-deoxy-D-glucose had only -2.25 kcal/mol that only met minimum affinity value (Reviewer-only Figure 2A).

To examine which drug candidate had shown the highest potential to bind with target proteins strongly and broadly, we set the threshold of lower 25% affinity values (affinity $<$ -7.81 kcal/mol) and counted the number of proteins per drug candidate (Reviewer-only Figure 2B). Cantharidin appeared to be the most interactive molecule, which was predicted to bind with 89 proteins, while 1,4-chrysenquinone, MG-132, and pararosaniline were predicted to interact with 81, 46, and 35 proteins, respectively. With the threshold of strong interaction as the affinity less than -10 kcal/mol was set roughly, 1,4-chrysenquinone, MG-132, and pararosaniline showed 5, 5, and 2 interactions, respectively, while no strong interaction was observed for cantharidin (Reviewer-only Figure 2C). Among these strongly predicted interactions, the minimum affinity was observed between the drug candidate MG-132 and target protein IFITM1 (affinity = -11.82 kcal/mol).

Through these additional analyses, we found that AD genes from our study and drug candidates have suggestive potential of physical interaction. The authors strongly believe that these results may offer the possibility to proceed with functional studies. We hope this additional work could address your comment well enough.

Reviewer-only Figure 2. The results of virtual docking simulation between potential drug candidates and 281 protein structures of TWAS-genes and meta-signatures.

(A) The boxplot showing the distribution of affinity values (free energy) for each drug candidate. (B) A bar plot presenting the number of interactions with the affinity < -7.81 kcal/mol. (C) A heatmap showing the affinity score between the strongly predicted pair of drug candidates and proteins (affinity < -10 kcal/mol).

Additionally, we designed functional experiments that may validate the efficacy of our drug candidates. In order to test the effects of the putative drugs *in vitro*, there are several models available, including 2D models to human skin equivalent, reconstructed human epidermis, and skin explants, even though none of them can exactly mimic human AD pathogenesis. Because we are simply trying to pre-screen drug candidates that would alleviate AD-like inflammatory symptoms, we would choose human HaCaT keratinocytes, because it is the most commonly used cells in the AD model. HaCaT cells produce various AD-related pro-inflammatory mediators upon several stimuli such as tumor necrosis factor (TNF)- α /IFN- γ .

First, we would conduct MTT assay using our putative drugs (pararosaniline, 2-deoxy-D-glucose, cantharidin, MG-132, and 1,4-chrysenequinone) to determine the maximum concentration at which cell survival rate is not affected. The reference drugs, tacrolimus, hydroxyzine, diphenhydramine, and cefalexin, would be used as positive controls and vehicle control (probably DMSO) as a negative control. HaCaT cells would be stimulated with 10 ng/ml of TNF- α /IFN- γ at 37°C with putative drugs, positive controls, and negative controls, respectively. Then the proteins would be extracted from cells and separated by SDS-PAGE and electroblotted onto an appropriate membrane. Then the membrane would be used for Western blot analysis, using β -actin as internal control and different primary antibodies against cytokines and chemokines. The primary antibodies would include but not limited to NF- κ B p65, p-Akt, p-STAT1 (Tyr701), IgE, p-STAT1 (Ser727), STAT1, p-I κ B- α , I κ B- α , Akt1/2/3, PARP.

Then mRNA would be extracted from the HaCaT cells treated with potential drugs, positive control, and vehicle control, respectively, and RT-qPCR would be conducted using primers for *LINGO4*, *RBM17*, *P4HA2*, *C1orf162*, *NOCT*, *TIGAR*, *SCIN*, and *BOC* which are

our novel genetic markers from skin tissue for the pathogenesis of AD, and *TSLP*, *TARC*, and *RANTES*, which are also known to play active roles in the pathogenesis of AD. mRNA expression levels of Th2 cytokines (*IL-4*, *IL-5*, and *IL-13*) and pro-inflammatory cytokines (*TNF- α* and *IL-6*) would also be measured.

If our drug candidate seems to be effective on $\text{TNF-}\alpha/\text{IFN-}\gamma$ -induced HaCaT keratinocytes, we can proceed with the *in vivo* validation using AD murine models. The initial dose for each drug candidate can be determined by using the lethal dose 50% (LD_{50}) values or the maximum available dosage that did not cause toxicity from previous *in vivo* studies (Reviewer-only Table 2). If both information is not available, we may start from 10% of the LD_{50} concentration of the reference drug that was predicted to show the most similar MOA with the drug candidate. Because it was hard to find the LD_{50} or available dosage value for 1,4-chrysenequinone, initial dose for 1,4-chrysenequinone need to be determined by using the LD_{50} value of the most nearly connected reference drug. Directly connected reference drug with 1,4-chrysenequinone was cefalexin and its oral LD_{50} dose in rats is 5,000mg/kg. Hence, the initial dose for 1,4-chrysenequinone may be set to 500mg/kg for rats and its dosage for mouse models are 1,000mg/kg based on the weight-body surface area ratio. The maximum range would be set to a point between no-observed-adverse-effect level (NOAEL) to LD_{50} . We would look for the signs of ameliorating AD-like symptoms within this range by using H&E stain histology, immunofluorescence, and Western blot analysis.

Even though none of them can exactly mimic human AD pathogenesis, there are several AD animal models that are available to observe the amelioration of AD-like symptoms, such as hapten-induced mouse models, ovalbumin models, and the MC903 models, epidermal barrier disruption and epicutaneous sensitization to common allergens, engineered mouse mutants, humanized mouse models. Assuming our drug similarity analysis

has provided reliable results, we expect 2 drug candidates (2-deoxy-D-glucose and pararosaniline) to relieve the itchiness, 2 other candidates (1,4-chrysenequinone and cantharidin) to prevent secondary infection of AD-like lesions, and MG-132 may show both effects moderately.

Addressing this issue, we realized that the statement from our original manuscript was too much aggressive (12p, line 24 – 13p, line 1). Although the genetic markers from our analysis need to be validated with human AD patients because of the genetic and transcriptomic differences between human and murine models; however, efficacy of the drug candidates can be evaluated enough by using the AD-like cell-line and murine models. Still, *in vitro* or *in vivo* works require massive manpower, time and equipment for experimental procedure, we thought this works need to be carried out by multiple follow-up studies. We believe that our study is still notable to induce the researchers in relevant fields to conduct deeper research using the results of our analysis. Therefore, we corrected the statement as below (14p, line 11-15).

Transcriptomic profiles of each murine model with AD-like phenotypes showed significant differences from human AD, indicating that our genetic markers need to be validated in human patients.⁹ However, since our drug candidates are associated with ameliorating the symptoms of AD, the effectiveness could be validated using *in vitro* and *in vivo* models

Reviewer-only Table 2. The list of the toxicity measurements (LD₅₀) of the drug

candidate molecules. If the exact LD₅₀ values were not founded, available dose without toxicity was searched from previous publications.

Drug Candidate	Experiment	LD₅₀ / Available dose
Pararosaniline	Mouse, oral	5.0 (mg/kg)
2-deoxy-D-glucose	Mouse, intravenous	8,000 (mg/kg)
Cantharidin	Mouse, intraperitoneal	1.0 (mg/kg)
MG-132	Mouse, intraperitoneal	10 (mg/kg) (Available dose)
1,4-chrysenequinone	-	-

Minor comments

Comment 1. How many genes were used in the PCA of Figure 2A?

>>> Response 1: All 11,965 genes available for meta-analysis were used for conducting PCA. We modified the figure legend of Figure 2A as below.

A scatter plot displaying the PCA results using all 11,965 genes after the batch effect correction.

Comment 2. The P-value of Pearson's correlation coefficient is 0 or any value are very small? Can you give an accurate value?

>>> Response 2: We agree that stating accurate P-value instead of rounded value will better present our results. The accurate P-value of Pearson's correlation coefficient was 2.791×10^{-11}

and this value is updated in the modified figure as below (Figure 4B and Reviewer-only Figure 3).

Reviewer-only Figure 3. The modified version of Figure 4B

Comment 3. Are there any small molecules in the CMAP database that have been used for AD treatment? What are their enrichment scores?

>>> The 4 reference drugs used for our comparison (diphenhydramine, tacrolimus, hydroxyzine, and cefalexin) were all stored in the CMAP database; however their enrichment scores were ambiguous. Only cephalexin showed consistent positive enrichment in the potential amelioration of AD transcriptional changes. The results are listed in the Reviewer-only Table 3.

Reviewer-only Table 3. CMAP enrichment results of reference AD drugs

Enrichment profiles of 4 reference AD drugs used in this study. Enrichment scores calculated with TWAS statistics and meta-analysis results are listed.

Cmap Name	Enrichment-TWAS	Enrichment-Meta
Diphenhydramine	-0.554	-0.561
Tacrolimus	-0.450	-0.684
Hydroxyzine	-0.443	0.266
Cefalexin	0.418	0.403

Comment 4. The author found 4 and 5 novel AD-related genes through TWAS and meta-analysis respectively. Compared with known genes, what new biological implications can these genes provide for our understanding of AD?

>>> The authors agree with the reviewer's comment that the new biological implications suggested by the novel genes need to be emphasized. The known AD genes represent the key biological pathways such as immune responses, skin barrier dysfunction, immune cell activations, and the novel genes from our work were all involved in these categories at pathway levels. In addition, we were able to observe the association between the hedgehog signaling pathway and AD in both TWAS and meta-analysis. Considering the recently emerging importance of the hedgehog signaling pathway in the irregular activity of T-cells in AD lesions, our data can support this recent point of view. Hence, we modified the sentence in the discussion section of the revised manuscript to describe this implication specifically.

(12p, line 21-24)

The pathogenetic role of hedgehog signaling in AD has received some attention in recent experimental studies, and our study also revealed the connection between AD etiology and the abnormal activation of this signaling pathway.

>>> Also, we feel that summarizing the biological mechanisms of the novel genes will be helpful to magnify our findings. The following sentence was modified as below in our revised version of the manuscript. (14p, line 6-9)

We identified novel genetic factors associated with AD risk and/or pathogenesis, which have roles in skin barrier abnormality, immune cell dysregulation, cell cycles, and immune responses, through an integrative transcriptome approach.

Comment 5. The x-axis of supplementary figure S2 is unclear.

>>> The authors are truly thankful for correcting our mistakes. The x-axis represents the genes involved in the locus. This description was added in our revised version of the supplementary figure file.

Comment 6. What is the threshold for differentially expressed genes in transcriptome meta-analysis?

>>> The authors are grateful for the helpful comments on our work. We found that the threshold for meta-analysis was only mentioned in the results section, not in the methods section. We agree with the reviewer that this may confuse the readers who seek for the threshold in the methods section. To address this issue, we also mentioned the threshold for meta-analysis in the methods section of the revised version of the manuscript. Again, thank you very much for the helpful comments to improve our work.

References

- 1 Ochoa, D. *et al.* Open Targets Platform: supporting systematic drug-target identification and prioritisation. *Nucleic Acids Res* **49**, D1302-D1310, doi:10.1093/nar/gkaa1027 (2021).
- 2 Rappaport, N. *et al.* MalaCards: an amalgamated human disease compendium with diverse clinical and genetic annotation and structured search. *Nucleic Acids Res* **45**, D877-D887, doi:10.1093/nar/gkw1012 (2017).
- 3 Trott, O. & Olson, A. J. AutoDock Vina: improving the speed and accuracy of docking with a new scoring function, efficient optimization, and multithreading. *J Comput Chem* **31**, 455-461, doi:10.1002/jcc.21334 (2010).
- 4 Forli, S. *et al.* Computational protein-ligand docking and virtual drug screening with the AutoDock suite. *Nat Protoc* **11**, 905-919, doi:10.1038/nprot.2016.051 (2016).
- 5 Varadi, M. *et al.* AlphaFold Protein Structure Database: massively expanding the structural coverage of protein-sequence space with high-accuracy models. *Nucleic Acids Res* **50**, D439-D444, doi:10.1093/nar/gkab1061 (2022).
- 6 Jumper, J. *et al.* Highly accurate protein structure prediction with AlphaFold. *Nature* **596**, 583-589, doi:10.1038/s41586-021-03819-2 (2021).
- 7 Aydin, A. D., Altinel, F., Erdogmus, H. & Son, C. D. Allergen fragrance molecules: a potential relief for COVID-19. *BMC Complement Med Ther* **21**, 41, doi:10.1186/s12906-021-03214-4 (2021).
- 8 Ortiz, C. L. D., Completo, G. C., Nacario, R. C. & Nellas, R. B. Potential Inhibitors of Galactofuranosyltransferase 2 (GlfT2): Molecular Docking, 3D-QSAR, and In Silico ADMETox Studies. *Sci Rep* **9**, 17096, doi:10.1038/s41598-019-52764-8 (2019).
- 9 Ewald, D. A. *et al.* Major differences between human atopic dermatitis and murine models, as determined by using global transcriptomic profiling. *J Allergy Clin Immunol* **139**, 562-571, doi:10.1016/j.jaci.2016.08.029 (2017).

REVIEWERS' COMMENTS:

Reviewer #3 (Remarks to the Author):

The authors have addressed the questions.